# Photoswitchable paclitaxel-based microtubule stabilisers allow optical control over the microtubule cytoskeleton

Adrian Müller-Deku [1], Joyce C. M. Meiring [2], Kristina Loy[1], Yvonne Kraus[1], Constanze Heise[1], Rebekkah Bingham[1], Klara I. Jansen[2], Xiaoyi Qu[3], Francesca Bartolini[3], Lukas C. Kapitein[2], Anna Akhmanova [2], Julia Ahlfeld [1], Dirk Trauner [4,5] & Oliver Thorn-Seshold [1,5✉]

Small molecule inhibitors are prime reagents for studies in microtubule cytoskeleton research, being applicable across a range of biological models and not requiring genetic engineering. However, traditional chemical inhibitors cannot be experimentally applied with spatiotemporal precision suiting the length and time scales inherent to microtubule-dependent cellular processes. We have synthesised photoswitchable paclitaxel-based microtubule stabilisers, whose binding is induced by photoisomerisation to their meta-stable state. Photoisomerising these reagents in living cells allows optical control over microtubule network integrity and dynamics, cell division and survival, with biological response on the timescale of seconds and spatial precision to the level of individual cells within a population. In primary neurons, they enable regulation of microtubule dynamics resolved to subcellular regions within individual neurites. These azobenzene-based micro-tubule stabilisers thus enable non-invasive, spatiotemporally precise modulation of the microtubule cytoskeleton in living cells, and promise new possibilities for studying intracel-lular transport, cell motility, and neuronal physiology.

[1] Department of Pharmacy, Ludwig-Maximilians University, Butenandtstrasse 5-13, Munich 81377, Germany. [2] Cell Biology, Neurobiology and Biophysics, Department of Biology, Faculty of Science, Utrecht University, Padualaan 8, 3584 Utrecht, The Netherlands. [3] Department of Pathology & Cell Biology, Columbia University Medical Center, New York, NY 10032, USA. [4] Department of Chemistry, New York University, 100 Washington Square East, New York, NY 10003, USA. [5]These authors jointly supervised this work: Dirk Trauner, Oliver Thorn-Seshold. ✉email: oliver.thorn-seshold@cup.lmu.de

The cytoskeleton serves as the scaffold for critical biological processes ranging from signalling and cargo trafficking, to cell shape maintenance and cell division. Most of the cytoskeleton's myriad of biological roles are inherently spatially and temporally differentiated, although they all rely on the same protein scaffold structures. Studying these cytoskeleton-dependent processes with the spatiotemporal resolution necessary to understand and exploit these individual biological functions is an important challenge. Accordingly, in recent years, a number of approaches towards photocontrol over cytoskeletal or cytoskeleton-associated proteins have been made, aiming to allow their spatiotemporally precise optical manipulation[1–3]. Nevertheless, much remains to be done before generally applicable tool systems can be developed.

We here focus on the microtubule (MT) cytoskeleton. MTs play particularly important roles in intracellular transport, cell motility and morphological plasticity, and there is a conspicuous need to achieve a better understanding of how these many functions are implemented and regulated[4,5]. The role of MT dynamics during cell proliferation has also made them a major anticancer target, for which several outstanding drugs (taxanes, epothilones and vinca alkaloids) have been developed[6–8]. These drugs and other small molecule modulators (e.g., nocodazole, combretastatin and peloruside) remain the most general tools for MT cytoskeleton research. However, these inhibitors simultaneously suppress all MT-dependent functions spatially indiscriminately. Therefore, they do not allow spatiotemporally precise MT inhibition on the length or time scales appropriate for selectively studying MT-dependent processes. This restricts their scope of applications and their utility for selective research into MT cytoskeleton biology[9].

Deeper insights could be gained from inhibitors that allow spatiotemporally specific MT manipulation. In this regard, optogenetic approaches to MT regulation have advanced greatly in recent years. In one example, a photo-inactivatable variant of the MT plus tip adaptor protein EB1 was engineered, which upon illumination lost its ability to bind other plus-end-tracking proteins, thus optically inhibiting MT growth and allowing spatiotemporally resolved modulation of directional cell migration[10]. Another domain of research has focused on optogenetic switches tethered to kinesins and compartment identity markers, to manipulate the transport of vesicles and organelles. With such methods, it is possible to photomodulate the association of motor proteins to cargos, to investigate the role of MTs in specific transport processes[11,12].

However, while optogenetics has succeeded in providing motors and scaffold-associated proteins that are responsive to externally controlled stimuli, no optogenetic variants of the basic cytoskeleton scaffold proteins actin and tubulin have been achieved. An exogenously controllable system for directly patterning cytoskeleton scaffold dynamics and structure with spatiotemporal resolution would however be highly desirable, since it would allow researchers to modulate any of the cytoskeleton-dependent functions. For this purpose, pharmacological interventions that directly address the stability and dynamics of the cytoskeleton scaffold remain the methods of choice. For example, photouncageable versions of MT inhibitors, particularly of the blockbuster drug paclitaxel, have been used for localised photo-activatable inhibition of MT dynamics in several studies[13,14]. Yet despite their ability to modulate the cytoskeleton directly, photouncaging approaches suffer disadvantages, such as irreversibility of inhibition, the intense and phototoxic mid-UV illumination typically needed for uncaging, the often slow and rate-limiting intermediate hydrolysis that diminishes their precision of temporal control, their release of nonspecifically toxic and also phototoxic photouncaging byproducts, background

activity through enzymatic hydrolysis, and increased molecular weight that can cause biodistribution problems[15].

Against this background, photopharmaceuticals—photoswitchably potent exogenous small molecule inhibitors—have been extensively developed in recent years[16–19]. Photopharmaceuticals conceptually enable studies not otherwise accessible to biology, marrying the spatiotemporal precision of light application known from optogenetics, to the flexibility and system-independence of exogenous small molecule inhibitors. This combination is favourable for non-invasive studies of temporally regulated, spatially anisotropic biological systems—such as the MT cytoskeleton[16,19,20]. Photopharmaceuticals have succeeded in delivering a measure of optical control over a broad range of biochemical and biological phenomena, with early cell-free studies now supplanted by applications in cultured cells and recently in vivo in embryonic and adult animals[21–24].

In the cytoskeleton field, several photopharmaceutical MT destabilisers were recently developed, to begin addressing the need for spatiotemporally precise MT cytoskeleton studies. The azobenzene-based Photostatins (**PST**s), which can be reversibly photoswitched by low-intensity visible light between their biologically inactive *E*-isomers and their MT-destabilising colchicine-like *Z*-isomers, were first developed in 2014[16,25–27]. MT-destabilising photopharmaceuticals based on two different families of molecular photoswitch—styrylbenzothiazoles (**SBTubs**)[28] and hemithioindigos (bi-active **HOTubs**[29] and dark-active **HITubs**[30])—have since been developed, delivering increased metabolic robustness in the intracellular environment and alternative optical switching profiles (all-visible switching with hemithioindigos, GFP-orthogonal switching with **SBTubs**). All three reagent families have enabled spatiotemporally precise optical control over endogenous MT network integrity, MT polymerisation dynamics, cell division and cell death. **PST**s have already been used in animals to tackle unsolved questions in mammalian development[21,22] and neuroscience[31]. These applications illustrate the power of photopharmacology to enable previously inaccessible studies of spatiotemporally anisotropic cytoskeletal processes without genetic engineering.

With the optically precise destabilisation of MTs addressed by a range of agents, we desired to develop photopharmaceutical MT stabilisers as conceptually novel tools with an alternative spectrum of biological research applications. While both MT destabilisers and MT stabilisers can be used to suppress MT polymerisation dynamics in cell culture, stabilisers have enabled a variety of research and human therapeutic applications which are inaccessible to destabilisers[14,32,33], due to their differing pharmacology, stoichiometry and spectrum of biological effects. Since the biological functions of the MT cytoskeleton are primarily dependent on the localisation of stabilised or growing MTs themselves, we reasoned that optically controlled tools modulating MT network stability could allow spatiotemporally precise stimulation of a range of these MT-dependent functions, in ways not otherwise amenable to control.

We therefore chose to develop light-responsive paclitaxel analogues as optically controlled MT stabilisers for in situ spatiotemporally precise photocontrol of cellular MT network architecture, dynamics and MT-dependent functions. We now report our development of these reagents.

## Results

**Design and synthesis.** We chose the azobenzene photoswitch for installing photoswitchable potency onto the taxane core. This photoswitch offers a substantial geometric change upon isomerisation, which we hoped would differentiate the isomers' binding constants, and it allows reliable, high-quantum-yield,

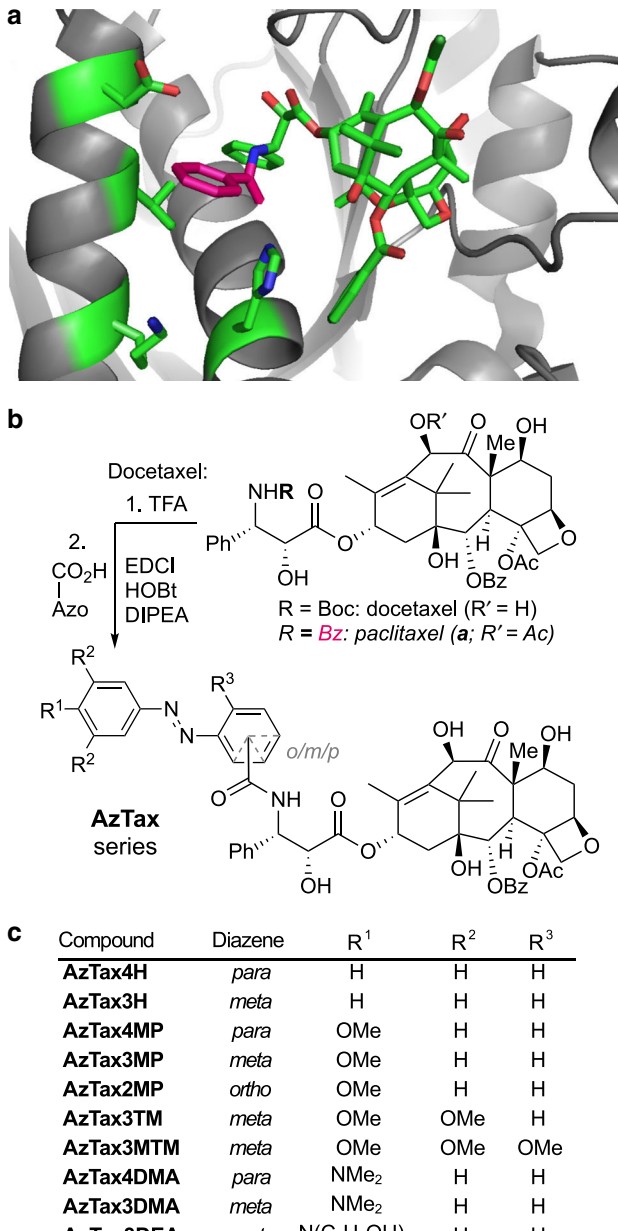

**Fig. 1 Design and synthesis of AzTax. a** Paclitaxel:tubulin structure (PDB: 3J6G[36]) with the benzamide indicated in pink. **b** Synthesis of **AzTax** from docetaxel. **c** Panel of **AzTax** examined in this study.

| Compound | Diazene | R¹ | R² | R³ |
|---|---|---|---|---|
| **AzTax4H** | *para* | H | H | H |
| **AzTax3H** | *meta* | H | H | H |
| **AzTax4MP** | *para* | OMe | H | H |
| **AzTax3MP** | *meta* | OMe | H | H |
| **AzTax2MP** | *ortho* | OMe | H | H |
| **AzTax3TM** | *meta* | OMe | OMe | H |
| **AzTax3MTM** | *meta* | OMe | OMe | OMe |
| **AzTax4DMA** | *para* | NMe₂ | H | H |
| **AzTax3DMA** | *meta* | NMe₂ | H | H |
| **AzTax3DEA** | *meta* | N(C₂H₄OH)₂ | H | H |

moderate potency loss[35], making it desirable for photo-pharmaceutical tuning as it might tolerate azobenzenes with a range of structural characteristics. However, we anticipated that attenuating the high potency of paclitaxel itself (low nM range) might be required, in order that the relatively small structural change of a *E/Z* isomerisation at the molecular periphery could substantially modify the overall potency.

We accordingly designed a panel of 3′-azobenzamide-taxanes (**AzTax**) for biological testing. As taxanes have famously poor aqueous solubility (still worsened by attaching an azobenzene), we initially determined to focus on compounds displaying satisfactory potency at concentrations substantially below their solubility limit. This avoids the case that the compounds' apparent potencies would be dictated by solubility effects, and so should enable robust use as reagents across a variety of systems and settings. Theorising that the sterics around the azobenzene phenyl ring proximal to the taxane core would be the greatest potency-affecting factor, we first focussed on testing which orientations of photoswitch would be best tolerated. We therefore scanned orientations of the diazene in *ortho*, *meta* and *para* relative to the amide (**AzTax2/3/4** compound sets, Fig. 1b, c), and when early cellular testing showed that the **AzTax2** set had the lowest potency, we abandoned it at this stage.

Next, examination of the published tubulin:paclitaxel cryo-EM structures (Fig. 1a)[36,37] indicated that the azobenzene's distal ring can project freely away from the protein. Therefore, we hypothesised that steric variation to the distal ring would not greatly impact binding potency of either isomer, but could be used orthogonally to tune their photochemical properties, by substitutions in *para* to the diazene that chiefly mesomerically affect the photochemistry of the N=N double bond. We accordingly synthesised unsubstituted ("H"), *para*-methoxy ("MP") and *para*-dimethylamino ("DMA") derivatives of the **AzTax3/4** sets. These were chosen to vary the photochemical properties of most relevance to photopharmacology: the completeness of the *E→Z* and the *Z→E* photoisomerisations at fixed wavelengths, which dictate the dynamic range of isomer photoswitchability, and *τ* (the halflife of the spontaneous unidirectional *Z→E* relaxation). Lastly, when the **AzTax3** set proved promising in early studies, we also examined installing an electron-donating 3,4,5-trimethoxy motif on the distal ring (**AzTax3TM**) as well as an additional R³ methoxy group to reduce the rotatability of the proximal ring in case this could amplify the difference between isomer potencies (**AzTax3MTM**), and we controlled for solubility effects by exchanging the dimethylamino substituent for a more soluble diethanolamino ("DEA") group (**AzTax3DEA**). The target **AzTax** were synthesised by degradation of commercial docetaxel followed by amide couplings to various azobenzenecarboxylic acids in moderate yields (Fig. 1b, c and Supplementary Note 1).

**Photochemical characterisation.** The **AzTax** all displayed robust and repeatable *E↔Z* photoswitching under near-UV/visible illuminations, as expected from the literature[19] (Supplementary Fig. 1). Since their protein target is located in the cytosol, we first wished to evaluate their photoswitching in physiological aqueous media. Since taxanes are too poorly water-soluble to perform reliable photoswitching studies easily by UV–Vis spectroscopy, we synthesised fully water-soluble diethanolamides of all the azobenzenecarboxylic acids and used them in aqueous photoswitching tests (see Supplementary Note 2). The photochemical properties within each substituent set were similar. The unsubstituted (H) compounds displayed a 3-fold dynamic range of *Z*-isomer photoswitchability between the photostationary states (PSSs) at 375 nm (80% *Z*) and 410 nm (26% *Z*), and had

near-UV/visible-light-mediated, highly robust *E↔Z* photo-isomerisability, which enables repeated photoswitching in situ in living cells. Taxanes feature a number of chemically modifiable positions; we chose to focus on sites where substituents can be tolerated, but where their geometric changes might impact binding potency through steric interactions or by modulating the orientation of key interacting groups nearby. Potent taxanes feature a side-chain 3′-amine acylatedsubstituted with mid-size hydrophobic groups (e.g., Boc group in docetaxel and Bz in paclitaxel)[8,34] which abut the tubulin protein surface yet are projected away from the protein interior (Fig. 1a, highlighted in pink); the other side-chain positions (e.g., the 3′-phenyl or 2′-hydroxyl) offer less tolerance for substitution as they project into the protein[8]. The 3′-amine also tolerates the attachment of somewhat polar cargos such as the large silarhodamine fluorophore, as long as they are attached via a long spacer, with only

substantially slower relaxation than biological timescales ($\tau$ ca. 50 days). The methoxylated compounds (MP, TM and MTM) had been chosen to improve the dynamic range of isomer photoswitching by relative shifting of the isomers' absorption bands[16] (Supplementary Fig. 2). Indeed, they delivered a ca. 9-fold dynamic range of $Z$-isomer photoswitchability (375 nm: 96% $Z$; 530 nm: 11% $Z$), and their relaxation remained substantially slower than biological timescales ($\tau$ ca. 24 h).

Advantageously for practical work, the metastable $Z$-isomers of all **AzTaxs** could be quantitatively relaxed to $E$ by warming DMSO stocks to 60 °C overnight (which increases the practical ease-of-use of these reagents as compared to irreversibly photouncaged reagents). The *para*-amino (DMA and DEA) compounds featured $\tau$ values too small to observe bulk photoswitching in aqueous media under biologically applicable conditions. Yet, since less water-solvated environments such as lipid vesicles, membranes, or protein-adsorbed states are likely intracellular localisations for hydrophobic taxane conjugates, we then determined their photochemistry in moderately polar aprotic media (EtOAc). Here, they were easily bulk-switchable ($\tau$ ca. 11 min), giving a 4-fold dynamic range of $Z$-isomer photoswitchability (410 nm: 91% $Z$; 530 nm: 21% $Z$) (further detail in Supplementary Note 2). As the **AzTax** reagents were intended for use with microscopy, we also examined photoswitching of all compounds over a broader range of wavelengths, to determine what dynamic range of isomer photoswitchability would be accessible in practice, with standard (405, 488 and 514 nm) or more exotic (380, 440 and 532 nm) microscopy laser wavelengths (Fig. 2a, Supplementary Fig. 2 and Supplementary Table 1).

We then proceeded to explore the biological applicability of **AzTax** as photoswitchable MT stabilisers in living cells. Since near-UV illumination gave PSSs with high-$Z$ populations for all photoswitches, while thermal relaxation and maintenance in the dark returned the $E$-isomer quantitatively, we began by comparing all-$E$ "dark" conditions (all-$E$ stock applied, then maintained dark) with mostly $Z$ "360 nm" lit conditions (all-$E$ stock applied, then photoisomerised *in situ* by pulsed illuminations with low-power 360 nm LED light, giving a mostly-$Z$ PSS), to determine which structures allowed the highest fold difference of bioactivity.

**AzTax display photocontrolled bioactivity in living cells.** Since stabilisation of MTs in cells over a prolonged period blocks cell proliferation and ultimately causes cell death[8], we first assayed the **AzTax** for light-dependent cellular activity by the resazurin cell proliferation/viability assay. Viability dose-response curves under dark or UV conditions were assessed in the HeLa cervical cancer cell line (Fig. 2b, c). All compounds displayed dose-response curves with similar Hill coefficients as the parent drug docetaxel (Supplementary Fig. 3), which is in line with the conjecture that they act through the same mechanism, albeit with different potency. All compounds except the fast-relaxing **AzTax3DMA** had $Z$-isomers that were more potent, or else equipotent, to the $E$-isomers, suggesting that this trend in isomer-dependent cellular bioactivity across several photoswitch types has robust significance. $E$-**AzTax2MP** had the poorest overall potency (EC50 ca. 7 μM, all-$E$), which we took as indicating the unsuitability of *ortho* substitutions that likely project the azobenzene into solution (c.f. Fig. 1a) where its hydrophobicity could interfere with binding stability. By contrast, the **AzTax4** set featured compounds up to 100 times more potent, and structure-dependently covered a 40-fold potency range. However, despite the good isomeric photoswitchability of e.g., **AzTax4MP**, none of the **AzTax4** set displayed substantial *photoswitchability of bioactivity* (fold difference between the 360 nm and the dark bioactivity). We

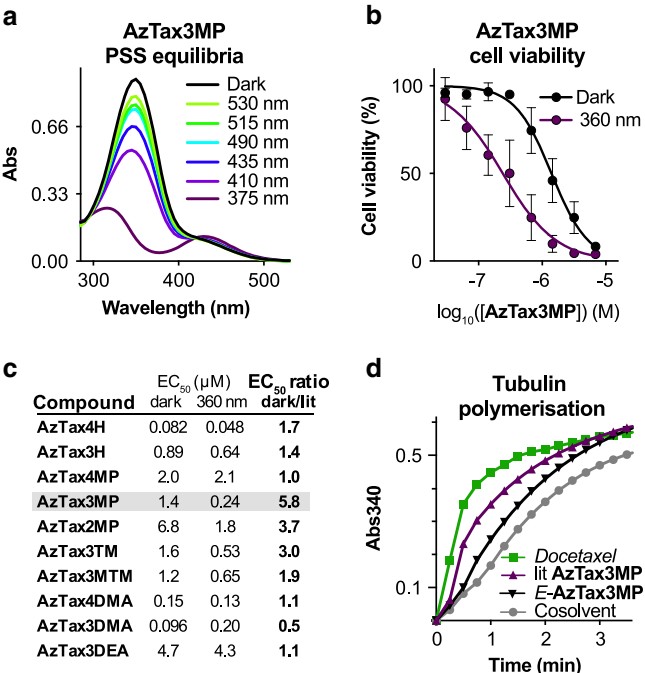

**Fig. 2 Photoswitchable performance of AzTax. a** Photostationary state UV–Vis absorption spectra of **AzTax3MP** under a range of cell-compatible wavelengths similar to microscopy laser lines. **b, c** Resazurin antiproliferation assays of **AzTax** highlight their structure- and light-dependent cell cytotoxicity. HeLa cells, 40 h incubation in dark conditions (all-$E$) or under pulsed illuminations with low-power LEDs (75 ms per 15 s near-UV at <1 mW cm⁻²; lit = ~80% $Z$) (dose-response curves are fitted to the means (shown with s.d.) of 3 independent biological experiments). **d** A cell-free assay for polymerisation of purified tubulin comparing the MT stabilisation activity of docetaxel, all-$E$- and 360 nm-lit-**AzTax3MP** (all 10 μM) shows light-specific promotion of polymerisation by $Z$-**AzTax3MP**, matching the trend observed in cellular assays.

interpreted this substitution-independent result as an indication that the distal ring may project too far from the protein contact surface, in both $E$- and $Z$-isomers, for the **AzTax4** isomer state to substantially affect binding.

In contrast, members of the **AzTax3** series all showed photoswitchability of bioactivity. **AzTax3MP** featured a nearly 6-fold difference between the more-toxic $Z$ and less-toxic $E$-isomers' bioactivity, over the 48 h experimental time course (Fig. 2b). Adding more methoxy groups to the scaffold decreased the $Z$-isomer's cytotoxicity without greatly affecting that of the $E$-isomer (**AzTax3TM**, **AzTax3MTM**), and deleting the methoxy group also decreased the $Z$-isomer's cytotoxicity (**AzTax3H**), which we took as a sign that balancing the polarity of the photoswitch was important for maximising bioactivity. In line with this interpretation, the potencies of **AzTax3DMA** were similar to **AzTax4DMA** while the more hydrophilic **AzTax3DEA** showed a 40-fold loss of potency. Despite the potential for photoswitching the *para*-amino **AzTax** inside lipid environments, they should only reach their cytosolic target tubulin as the $E$-isomers due to fast aqueous relaxation. Surprisingly, **AzTax3DMA** appeared slightly more bioactive as the unilluminated $E$-isomer, although as expected **AzTax4DMA** and **AzTax3DEA** both showed no illumination-dependency of bioactivity; and controls under 410 nm illumination (to establish the optimum PSS for the *para*-amino compounds) showed no different result to those obtained with UV illumination (Fig. 2). The apparent cytotoxicity differential seen for **AzTax3DMA** might reflect reduced availability to the cytosol rather than

differential binding of the isomers[30] although these results do not allow further conjecture.

To continue the study, we therefore selected **AzTax3MP**, due to its photoswitchability of bioactivity (6-fold when under optimum illumination conditions), satisfactory potency ($EC_{50} = 0.24\,\mu M$ when UV illuminated), bidirectional photoswitchability (optimum 9-fold-change of concentration of the more bioactive $Z$-isomer), and reproducibly photoswitchable cellular performance across assays with different illumination conditions, to proceed with further mechanistic biological evaluations.

**Photocontrol of tubulin polymerisation and cell cycle**. To examine the molecular mechanism of **AzTax** isomer-dependent cellular bioactivity, we first assayed the potency of **AzTax3MP** for promoting tubulin polymerisation in cell-free assays using purified tubulin. The majority-$Z$ 360 nm-lit state gave a ca. 60% enhancement of polymerisation over control (benchmarked to docetaxel at 100%), while all-$E$-**AzTax3MP** gave only ca. 30% polymerisation enhancement (Fig. 2d and Supplementary Note 2). Note that the enhancements seen in this cell-free assay are not predictive of the relative cellular potencies, but it clarifies the mechanism of action of **AzTax** as MT stabilisers, like their parent taxanes. We next studied the direct effects of in situ-photoisomerised **AzTax** upon cellular MT organisation and MT-dependent processes. Immunofluorescence imaging of cells after 24 h of exposure revealed that **AzTax3MP** light-dependently disrupts MT network architecture. As **AzTax** concentration increases, MTs first become disorganised, then mitotic spindle defects that result in multinucleated cells are seen, and finally mitotically arrested and early apoptotic cells with fragmented nuclei dominate; the effective concentrations needed to achieve these effects are substantially lower for lit **AzTax** than under all-$E$ dark conditions (Fig. 3, further detail in Supplementary Note 4). The best window for visualising this isomer-dependent bioactivity lay around 0.3–1 μM. Both the mitotic arrest, and the nuclear defects of cells that escape arrest, are hallmarks of MT stabiliser treatment[38], arguing that the isomer-dependent cytotoxicity of **AzTax3MP** arises from MT stabilisation preferentially by its $Z$-isomer.

The presence of multinucleated cells indicated that **AzTax** also inhibit MT-dependent functions such as successful completion of mitosis. To quantify this we examined cell-cycle repartition after **AzTax** treatment by flow cytometry, expecting to observe $G_2/M$-phase cell-cycle arrest[39]. $G_2/M$-arrest was observed with approximate $EC_{50}$ around 1.5 μM after 24 h incubation with the lit **AzTax3MP**. This was twice as potent as $E$-**AzTax** (Fig. 4a–c) and mimicked the effect of docetaxel although with lower potency (Supplementary Fig. 4). As a control for illumination/photoswitch-dependent off-target effects, we also examined the non-photoswitchably bioactive but potent **AzTax4DMA**, which reproduced the effects of docetaxel independent of illumination conditions indicating no significant assay complications (Supplementary Fig. 4). This further supported the notion that **AzTax3MP** acts across a range of assays and readouts as a light-modulated taxane, with reproducible photocontrol over the isomers' bioactivity, allowing effective inhibition both of MTs and of MT-dependent processes.

**AzTax enable spatiotemporally specific MT control in cells**. With their photoisomerisation-dependent bioactivity shown in long-term experiments (days), we next tested one of the key conceptual advantages that photoswitchable MT stabilisers should enable: real-time in situ optical control of the MT cytoskeleton with single-cell spatial specificity and high temporal precision.

We transfected HeLa cells to express the EB3-tdTomato fluorescent reporter, that selectively labels GTP:tubulin-rich regions of MTs. In ordinary conditions this is a marker for the GTP cap region of polymerising MTs, thus selectively revealing polymerising MTs as comets moving towards the cell periphery[40]. As taxane treatment suppresses MT polymerisation dynamics[41], we imaged EB3-tdTomato dynamics using live cell confocal microscopy, as a spatiotemporally resolved readout for MT inhibition by **AzTax**.

We applied **AzTax3MP** globally to these cells at 1 μM, and targeted single cells with low-intensity illuminations at 405 nm, hoping to achieve precisely temporally resolved control of MT dynamics in those single cells only, by selectively isomerising **AzTax3MP** inside them. In targeted cells, EB3 comet counts were halved upon 405 nm illuminations, recovering almost completely within 80 s after 405 nm illumination was stopped (Fig. 5a), while surrounding cells as well as non-treated controls were unaffected (Fig. 5b); and the process could be repeated over several cycles (Supplementary Movies 1 and 2, Supplementary Note 4 and Supplementary Discussion). Thus, **AzTax** can indeed be used to reversibly inhibit MT dynamics of target cells with temporal precision on the scale of seconds and spatial precision on the level of tens of microns. To examine $E$-**AzTax3MP** (dark state) for residual effects on MT dynamics, we also quantified other parameters of EB3 motility. $E$-**AzTax3MP** caused only a slight decrease in EB3 comet velocities (Fig. 5c), and an insignificant decrease in EB3 comet density (Fig. 5d). By comparison, in photoswitched cells, so few comets remain that meaningful velocity quantification was not possible (Fig. 5c, d). It can be concluded that working concentrations can be appropriately chosen such that **AzTax3MP** does not hinder MT polymerisation in the absence of illumination, but does so strongly when illuminated.

Finally, to study the entire cellular MT population and not only the actively polymerising fraction of MTs, HeLa cells transfected to express mCherry-α-tubulin were imaged to quantify total MT density under **AzTax3MP** treatment (Supplementary Movies 3 and 4). No change in MT density was observed during treatment or photoswitching (Supplementary Fig. 5). Taken together, this argues that, as expected, **AzTax** do not act by reducing the overall number of MTs, but rather by suppressing MT polymerisation dynamics.

**AzTax allow subcellular photocontrol of neuronal MT dynamics**. Having demonstrated MT control with spatial resolution to single cells, we now wished to examine the performance of **AzTax** as applied to subcellular resolution. Small molecule inhibitors diffuse rapidly within cells, so using them to achieve sustained subcellular patterning of biological effects is challenging. Yet, for potential applications in neurobiology, the highly polarised, elongated neuronal cell shape should restrict diffusion and favour subcellular resolution. We used mature cultured rat hippocampal neurons with well-developed dendrites, and transfected these cells with EB3-tdTomato for imaging. For each neuron, we selected equal-sized areas along independent processes, and monitored their EB3 dynamics before and during **AzTax3MP** treatment, with ROI-localised 405 nm application to one area (Fig. 6 and Supplementary Movies 5–8).

Kymographs of these areas reveal localised reductions in MT polymerisation dynamics in 405 nm-illuminated ROIs in the presence of **AzTax3MP**, while no significant changes are seen in other dendritic processes; and internal controls before/after **AzTax** application confirmed that no significant photobleaching is caused by 405 nm illumination before **AzTax** application (Fig. 6a). The induction of inhibition in the ROI was typically

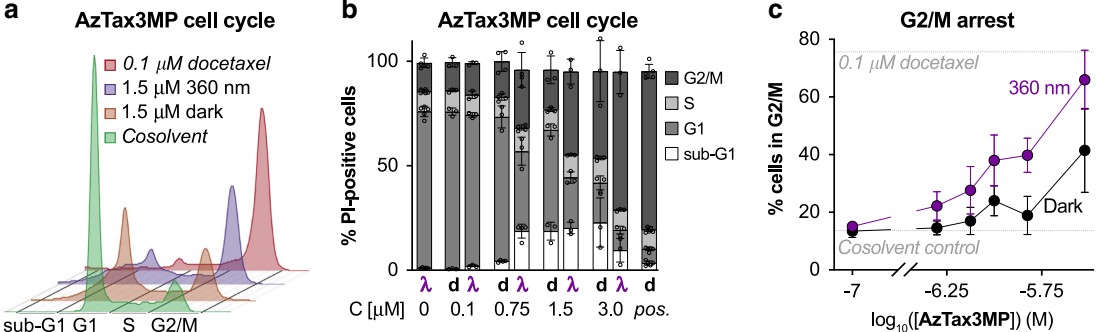

**Fig. 3 AzTax3MP light-dependently disrupts MTs in cultured cells.** Immunofluorescence staining indicates dose- and light-dependent disruption of MT organisation, mitotic completion, and cell viability. HeLa cells treated with **AzTax3MP**, docetaxel, or DMSO cosolvent only for 24 h; α-tubulin in green, DNA in blue; docetaxel positive control at 0.1 μM; DMSO cosolvent at 1% in all conditions; scale bars 25 μm (staining was repeated twice after two independent experiments and evaluated each time by at least two scientists independently).

**Fig. 4 AzTax3MP leads to dose- and light-dependent cell-cycle arrest. a** Flow cytometry analysis of cell-cycle repartition shows that **AzTax3MP** leads to a light-dependent shift of living cells towards $G_2/M$ phase. **b**, **c** The cell-cycle distribution of **AzTax**-treated cells approaches that of docetaxel-treated cells in a dose- and light-dependent manner. HeLa cells treated under light/dark conditions for 24 h (**d** indicates dark and **λ** indicates lit conditions; *pos.* indicates docetaxel control at 0.1 μM; $n = 3$ biologically independent experiments, except $n = 4$ for 0.75 μM **AzTax3MP** and $n = 5$ for cosolvent and docetaxel controls; data shown as mean with s.d.).

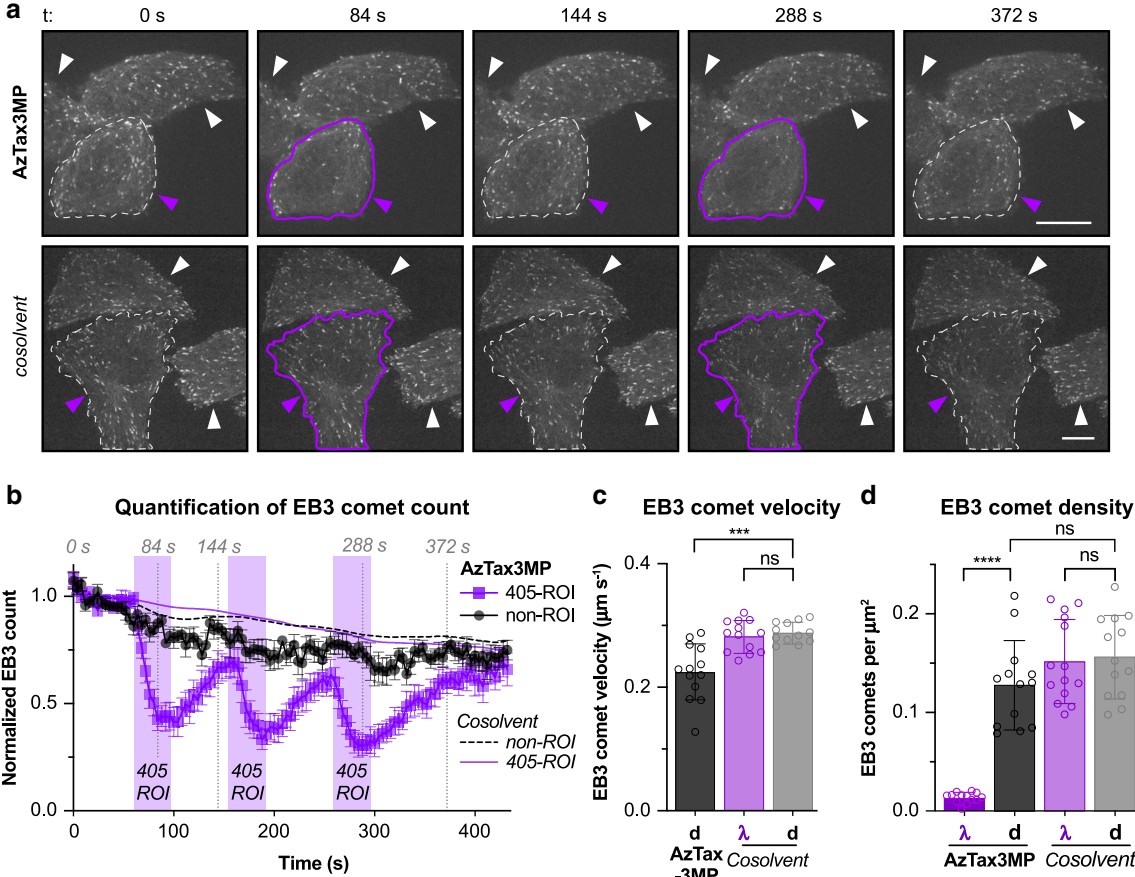

**Fig. 5 AzTax photoswitching allows cell-precise, temporally reversible inhibition of MT polymerisation dynamics.** Data related to Supplementary Movies 1 and 2. HeLa cells transfected with EB3-tdTomato treated with 1% DMSO cosolvent with or without 1 μM **AzTax3MP**. **a, b** Target cells (violet arrowheads) were selectively illuminated with 405 nm in bursts, and EB3 tracking in defined regions of interest (ROIs) performed. **a** Target ROIs in frames acquired immediately after 405 nm illumination are outlined in violet, target ROIs in $t_0$ and post-recovery frames (ca. 80 s after 405 nm illumination periods) are outlined in dotted white; nontarget cells are indicated with white arrowheads; scale bars indicate 10 μm. **b** Quantification of EB3 comet counts (averages over $n = 7$ cells per treatment group across 3 experiments, each cell's EB3 comet count time course was normalised to the average of the first ten frames) in target ROI and in non-illuminated cells. Periods of ROI illumination with 405 nm are shaded in violet; times of frames from the representative movies shown in **(a)** are indicated with dotted lines annotated with the times, data shown as group-average of EB3 comet count time course (normalised to initial count) with s.e.m. For clarity, data for cosolvent-only areas are shown with LOWESS fits. **c, d** HeLa cells transfected with EB3-tdTomato were treated with either 1 μM **AzTax3MP** or DMSO cosolvent only, and imaged directly after drug application for 2.4 min with 405 nm illumination (signified by "λ") or without (signified by "d"). Supplementary Movies were analysed for **c** EB3 comet velocity ($n = 12$ cells for **AzTax3MP**; $n = 13$ cells for cosolvent conditions), and **d** EB3 comet counts, normalised to cell surface area ($n = 13$ cells for **AzTax3MP** lit and both cosolvent conditions; $n = 14$ cells for **AzTax3MP** dark condition). EB3 comet velocity could not be meaningfully analysed in **AzTax3MP**-treated cells under 405 nm illumination due to the strong loss of EB3 comets and is therefore not represented. Data are shown as mean with s.d. Unpaired, two-tailed $t$ test: ***$P = 0.001$ (95% CI 0.035–0.093), ****$P < 0.0001$ (95% CI 0.089–0.14), ns denotes not significant, no adjustments for multiple comparisons.

clear within ca. 1–3 min, which is approximately one order of magnitude slower than in the whole-cell photoswitching experiments of Fig. 5, and which may partially reflect the lower **AzTax** concentration that was selected to avoid complications with the more sensitive neurons. Statistics collected over multiple cells highlighted the reproducibility of using ROI-localised **AzTax3MP** illumination to deliver subcellularly localised MT polymerisation inhibition in this system (Fig. 6b).

Collectively, these results demonstrate the use of **AzTax3MP** as a powerful tool to directly optically modulate endogenous MT network architecture, polymerisation dynamics and MT-dependent functions in live cells with excellent spatiotemporal control.

## Discussion

Photocontrol over protein function is an attractive method to study anisotropic, multifunctional cellular systems, since it can deliver the spatiotemporal specificity required to focus on specific roles or aspects within these complex biological systems. Small molecule photopharmaceuticals have already proven valuable photocontrol tools because of their ability to address such targets that are not directly accessible to optogenetics, such as the MT cytoskeleton (for which a range of photoswitchable depolymerising agents have recently been reported[16,28,30]). Here we have expanded the scope of photopharmaceutical MT reagents to demonstrate the first photoswitchable MT stabilising agents. Through structure-photochemistry/activity-relationship studies, we have selected a lead compound **AzTax3MP** that gives robust, in situ photoswitchable MT stabilising activity in cell-free and cellular assays, and can light-dependently reproduce key direct as well as downstream biological effects of the taxanes but with excellent spatiotemporal control down to the subcellular level. While this is a promising starting point for further reagent optimisation, we believe that **AzTax3MP** itself will already find a

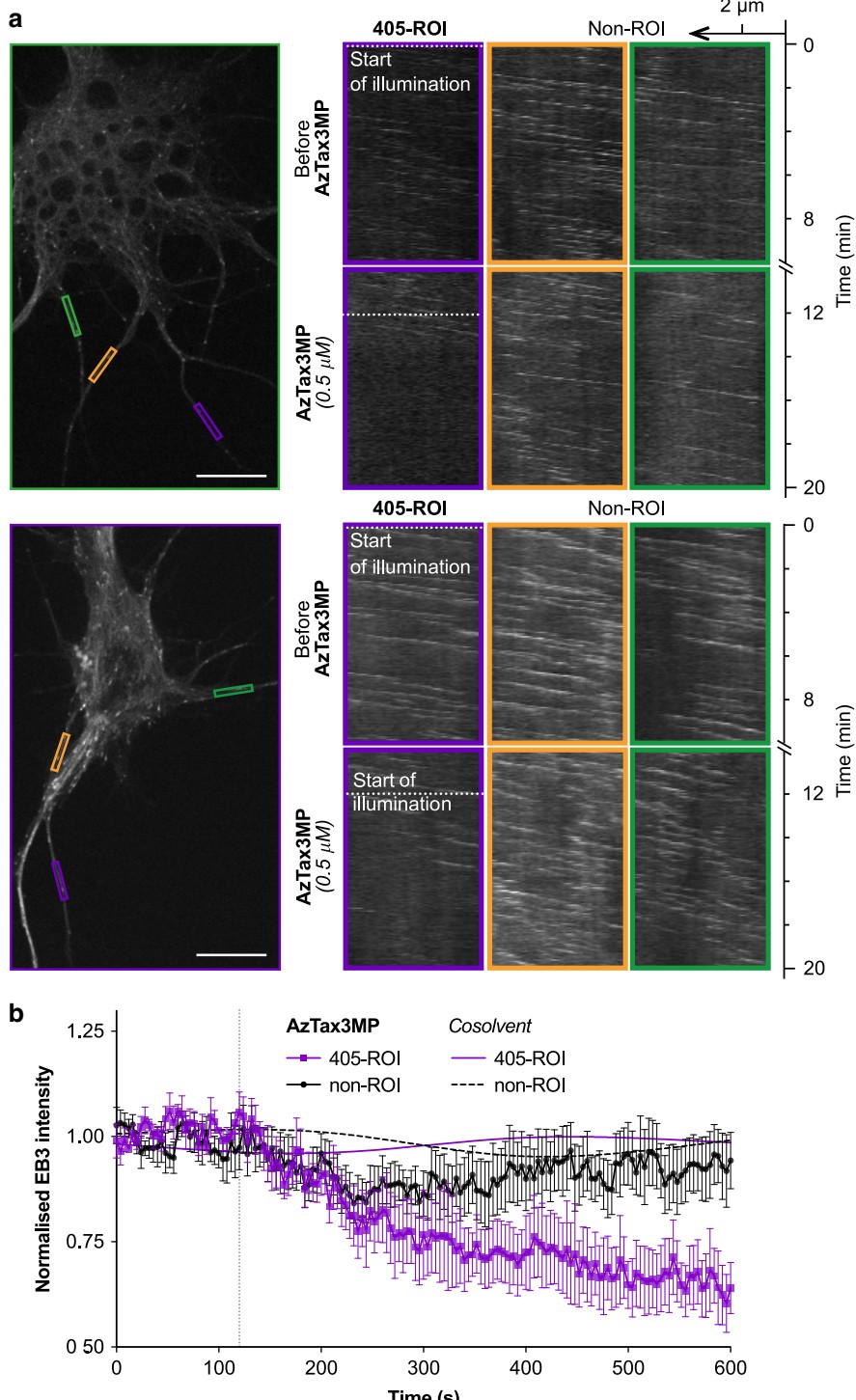

**Fig. 6 Manipulation of MT polymerisation dynamics in subcellular ROIs of rat primary hippocampal neurons using AzTax3MP.** Data related to Supplementary Movies 5–8. **a**, **b** Cultured primary neurons (9 days in vitro) transfected with EB3-tdTomato treated with 1% DMSO were initially imaged for EB3 for 10 min while a ROI (violet box) was pulsed with 405 nm light, establishing baselines for EB3 activity in the cell and in the ROI, which were demonstrated to be light-independent. The same neurons were then exposed to 0.5 μM **AzTax3MP** and immediately imaged for another 10 min; during this time the same ROI (violet box) was pulsed with 405 nm light beginning at 2 min into the acquisition (indicated by dotted lines). **a** Cell images with areas marked, and corresponding kymographs of these areas. The ROI pulsed with 405 nm is boxed in violet, the non-ROI areas (not pulsed with 405 nm) are boxed in orange and green. Scale bars indicate 10 μm. **b** Normalised area-average pixel EB3 intensities (with s.e.m.), for areas treated with or without **AzTax3MP** and 405 nm pulsing (*n* = 4 cells). For clarity, data for cosolvent-only areas are shown as spline fits; see Supplementary Note 4 for further details.

range of applications particularly in embryology, neuroscience and studies of cell motility and polarity, where its spatiotemporally specific bioactivity will enable studies not previously possible.

Liu et al. also recently reported photoswitchable MT stabilisation, using a photoswitchable supramolecular host-guest system intended to crosslink cellular MTs[42]. However, these effects were also reproduced when the tubulin stabilisation motif was deleted[43], and it remains unclear if the host-guest system could perform specific tubulin binding[44]. The druglike reagents developed in this work, offering robust and structurally rationalisable performance, may therefore be valuable to address still-unmet needs.

If **AzTax** reagents with still greater bioactivity differentials could be accessed, this would improve reagent performance with respect to several limitations of the current best candidate **AzTax3MP**. This reagent's dynamic range of bioactivity photoswitching is between two- and six-fold, depending on the assay readout. Thus, relatively precise concentration tuning is needed to find the best working concentration, and background activity before photoactivation may be observed depending on the assay and conditions. Determining the sources of the differential bioactivity between **AzTax** isomers in living cells is therefore key for reagent optimisation. Since modifying polarity at a distal site that should not clash sterically with the protein gave a 40-fold change of apparent potency (**AzTax3DEA** compared to **AzTax3DMA**), we believe that the sterics of the **AzTax** isomers are not necessarily the sole determinant of cellular bioactivity. Yet, polarity-dependent cellular biolocalisation or penetration cannot entirely explain the photoswitchable activity of **AzTax3MP** since it shows isomer-dependent activity in cell-free assays also, so the azobenzene must significantly impact protein-ligand affinity. Therefore, we conclude that maximising the bioactivity difference between isomers will require photoswitches with isomer-dependency both of sterics and of polarity. We note too that the completeness of the $E{\to}Z$ photoswitchability of the azobenzene was not correlated to the photoswitchability of biological activity (c.f. **AzTax3MTM**, **AzTax4MP**). Since the $Z$-**AzTax** were typically the more-active isomers, we conclude that further defavouring the binding of the $E$-isomer while allowing the $Z$-isomer to retain bioactivity, e.g. by tuning sterics and polarity, is likely the best way to maximise the photocontrol over inhibition. Research in these directions is underway.

Reducing the halflife of spontaneous relaxation of an **AzTax** to its less-binding isomer, could also offer perspectives for improving the ease of cellularly or subcellularly specific photocontrol. Faster relaxation could reduce the biological effects of a photoactivated isomer diffusing away from a desired spatial and temporal locality, and so improve spatiotemporal precision. However, if relaxation becomes too fast, the requirements for repeated localised illuminations may become the limiting factor for compound utility. We estimate that relaxation on the scale of seconds, or hundreds of microseconds, would best address most desirable short-term assays. While the relaxation speed of **AzTax3MP** was in this respect too slow, and that of **AzTax3DMA** was too fast, it is likely that by electronic tuning of the azobenzene's distal phenyl ring, more useful derivatives with intermediate relaxation rates can be obtained. Research in this direction is ongoing.

Photopharmacology studies often assume that increasing the dynamic range of isomeric photoswitchability under a freely available choice of illumination conditions, and red-shifting overall absorption wavelengths, are required for improved biological performance. However, in the case of the current or any potentially tuned future **AzTax** stabilisers, as elsewhere, probably neither is true:

First, after $Z$-**AzTax** induces increased MT rescues, which alter MT dynamics and network architecture, the downstream MT-dependent biology probably cannot be rapidly returned to its usual state even if the stabiliser would be totally removed (e.g., by complete back-isomerisation to a hypothetically non-binding state). This is because any **AzTax**-stabilised MTs are likely to be organised abnormally, and will presumably require time to break down and return tubulin monomer to the cytoplasmic pool so that a functional, directional MT network can be rebuilt. Therefore, there is probably a limit to the temporal resolution of *true* biological reversibility that any photoswitchable stabiliser can display, even if selected readouts (such as speed of polymerisation of individual MTs) recover more quickly. With this consideration in mind, we do not believe that improving the completeness of bidirectional isomeric photoswitchability[45] will be as important for **AzTax** development as for other classes of inhibitors that can feature instantaneous downstream biological response.

Second, red-shifting photoswitch absorption wavelengths is also likely to be counterproductive for microscopy, since there are few fluorescent proteins with significant excitation efficiency at laser lines above 561 nm (typically the next wavelength available is 647 nm). Therefore, maintaining orthogonality to the widest possible range of imaging wavelengths by blue-shifting is in our opinion and experience more advantageous, since it can keep other imaging channels vacant for multiparameter, photo-orthogonal studies[28]. However, a key property that should be readily tunable to the advantage of this system is the $E{\to}Z$ photoisomerisation efficiency at 405 nm, which is usually the only microscopy laser available in the 350–440 nm range. Here we consider that improved performance for **AzTax**-like reagents will depend on optimising photoconversion at the wavelength/s that will in practice be used for their photocontrol. Developing a set of standard photoswitches with better 405 nm $E{\to}Z$ photoconversion than these *para*-alkoxyazobenzenes (~46% $Z$) yet with similar polarity and substantial stability against thermal relaxation, is a nontrivial goal of our ongoing research.

From a biological perspective, while **AzTax** have reproduced the effects expected for taxanes across these studies, comparing their mechanism more deeply remains to be addressed. Given the structural overlap of the binding pharmacophores (Fig. 1; ca. 215 $\text{Å}^2$ of paclitaxel's total polar surface area (221 $\text{Å}^2$) overlaps with that of **AzTax3MP** (249 $\text{Å}^2$)) we consider it likely that their effects will prove similar in a variety of settings. This would open up possibilities towards e.g., time-resolved studies of the effects of taxane binding relying on these photoswitchable analogues.

The **AzTax** photoswitchable MT stabilisers can be used in conjunction with long term, in situ photoswitching in live cells to control critical biology from cytoskeleton architecture to cell survival. In short-term experiments, they can reliably apply cell-specific and temporally reversible MT inhibition under straightforward optical control, and have allowed subcellularly resolved inhibition in primary cells with unusual geometry. By complementing the existing MT-depolymerising photo-pharmaceuticals, the development of **AzTax** now brings both principal modes of pharmacological MT modulation under optical control.

**AzTax** reagents open up a multitude of possibilities for high-precision biological studies not possible with previous methods. They may contribute to studies across a variety of settings where temporally, cell- or subcellularly specific roles of MTs either are unclear, or else determine downstream biology that is itself of interest. For example, cell specific, temporally precise modulation of MT stability and dynamics may be particularly useful for studies of rapid coordinated processes such as mitotic progression, transport, migration and immune cell response, even in complex environments. **AzTax** may also find particular utility in

neurobiology, for example in exploring the recently discovered important roles of MTs in developing and regenerating neurons. Although MT stabilisation during development was shown to determine axonal identity and remodelling, MT stabilisation in mature neurons seems to promote axonal regeneration by reducing the formation of retraction bulbs and modulating glial scar formation after spinal cord injury[14,32,46–48]. However, the temporal characteristics of these phenomena are unclear, and the roles of MT stabilisation in surrounding glia and immune cells rather than in damaged neurons themselves have not yet been resolved. Photopharmaceutical stabilisers could shed new light on this field by stimulating the phenomena with high enough spatiotemporal resolution to clarify the primary processes responsible[32,33].

In the field of pharmacology, it is still to a large degree unclear[38] how the blockbuster taxane drugs exert their cellular/tissue-level effects in vivo. This provides enormous clinically driven interest in increasing the understanding of taxane pharmacology: both towards improved paclitaxel-site antimitotic therapeutics, and towards designing better combination treatment regimens involving these broad-spectrum cancer chemotherapeutics. Against this backdrop too, **AzTax** offer an intriguing method for precise studies, that may clarify the spatiotemporal dependency of biological action of the parent taxanes.

Beyond the first generation of **AzTax** reagents, we have also identified perspectives for further improving **AzTax**'s photocontrol of biological function through structure-photochemistry/activity-relationship studies. This opens up several avenues for applying fundamental research in the rapidly evolving field of chemical photoswitches to generate specialty MT stabiliser photopharmaceuticals for cell-free mechanistic studies, cell biology, and towards in vivo use. More broadly, this work will also guide and encourage further photopharmaceutical reagent development for other proteins inaccessible to direct optogenetics, including the actin cytoskeleton.

In conclusion, we believe that the **AzTax** will prove useful in studies from intracellular transport, cell division and cell motility, to neurobiology; and that this first demonstration of photoswitchable MT stabilisers is a substantial step towards high-spatiotemporal-precision studies of a range of critical processes in cell biology.

## Methods
Full and detailed experimental protocols can be found in the Supplementary Information.

**Compound synthesis and characterisation**. Reactions and characterisations were performed by default with non-degassed solvents and reagents (Sigma-Aldrich, TCI Europe, Fisher Scientific), used as obtained, under closed air atmosphere without special precautions. Manual flash column chromatography was performed on Merck silica gel Si-60 (40–63 μm). MPLC flash column chromatography was performed on a Biotage Isolera Spektra, using Biotage prepacked silica cartridges. Thin-layer chromatography was run on 0.25 mm Merck silica gel plates (60, F-254), with UV light (254 and 365 nm) for visualisation. NMR characterisation was performed on Bruker 400 or 500 MHz spectrometers. HRMS was performed by electron impact at 70 eV with Thermo Finnigan MAT 95 or Jeol GCmate II spectrometers; or by electrospray ionisation with a Thermo Finnigan LTQ FT Ultra Fourier Transform Ion Cyclotron resonance spectrometer. Analytical HPLC-MS was performed on an Agilent 1100 SL HPLC with H₂O:MeCN eluent gradients, a Thermo Scientific Hypersil GOLD™ C18 column (1.9 μm; 3 × 50 mm) maintained at 25 °C, detected on an Agilent 1100 series diode array detector (DAD) and a Bruker Daltonics HCT-Ultra mass spectrometer.

**Photocharacterisation**. UV–Vis-based studies (determination of absorption spectra, photostationary states, reversibility of photoisomerisation, and Z to E relaxation) were performed on a Varian CaryScan 60 (1 cm pathlength) at room temperature with model photoswitches that were water-soluble analogues of the **AzTax** species, since reliable UV–Vis studies require compound concentrations around 25–50 μM, while the **AzTax** compounds are only reliably molecularly soluble at such concentrations with high cosolvent percentages (e.g., 50%

DMSO) that do not reflect the intracellular environment and also alter the isomers' spectra, quantum yields, and relaxation times. We synthesised and used di(2-ethanol)amine carboxamides as water-soluble analogues of the taxane carboxamide **AzTax** (see Supplementary Information) enabling measurements in PBS at pH ~7.4 with only 1% of DMSO as cosolvent, thus matching the intracellular environment around the **AzTax**' protein target, tubulin. Star model 3W LEDs (360–530 nm, each FWHM ~25 nm, Roithner Lasertechnik) were used for photoisomerisations in cuvette that were thus predictive of what would be obtained in the cytosol during LED-illuminated cell culture. Spectra of pure E- and Z-isomers were acquired from the HPLC's inline Agilent 1100 series DAD over the range 200–550 nm, manually baselining across each elution peak of interest to correct for eluent composition.

**Tubulin polymerisation in vitro**. In total, 99% purity tubulin from porcine brain was obtained from Cytoskeleton Inc. (cat. #T240) and polymerisation assays run according to manufacturer's instructions. Tubulin was incubated at 37 °C with lit- or dark-**AzTax** (10 μM) in buffer (with 3% DMSO, 10% glycerol) and GTP (1 mM), and the change in absorbance at 340 nm was monitored over 15 min at 37 °C[49].

**Standard cell culture**. HeLa (ATCC CCL-2) and COS-7 (ATCC CRL-1651) cells were maintained under standard cell culture conditions in Dulbecco's modified Eagle's medium supplemented with 10% foetal calf serum (FCS), 100 U mL⁻¹ penicillin and 100 μg mL⁻¹ streptomycin, at 37 °C in a 5% CO₂ atmosphere (see also Supplementary Information for protocols for other cell types). For long-term assays under photoswitching, HeLa cells were transferred to phenol red free medium prior to assays. Compounds (in the all-E state) and cosolvent (DMSO; 1% final concentration) were added via a D300e digital dispenser (Tecan). Treated cells were then incubated under dark (light excluded) or lit conditions (where 75 ms illumination pulses were applied to microtiter plates every 15 s by self-built wavelength-specific multi-LED arrays, to create and maintain the wavelength-dependent photostationary state isomer ratios throughout the experiment[16]).

**Antiproliferation assay**. As a proxy readout for viable cells, mitochondrial diaphorase activity in HeLa cell line was quantified by measuring the reduction of resazurin (7-hydroxy-3H-phenoxazin-3-one 10-oxide) to resorufin. 5000 cells per well were seeded on 96-well plates. After 24 h, cells were treated with E-**AzTax**, shielded from ambient light with light-proof boxes, and exposed to the appropriate light regimes. Following 48 h of treatment, cells were incubated with 20 μL of 0.15 mg mL⁻¹ resazurin per well for 3 h at 37 °C. The resorufin fluorescence (excitation 544 nm, emission 590 nm) was measured using a FLUOstar Omega microplate reader (BMG Labtech). Results are represented as percent of DMSO-treated control (reading zero was assumed to correspond to zero viable cells) and represented as mean of at least three independent experiments with s.d.

**Cell-cycle analysis**. E-**AzTax** were added to HeLa cells in 6-well plates (seeding density: 300,000 cells per well) and incubated under dark or lit conditions for 24 h. Cells were harvested and fixed in ice-cold 70% ethanol then stained with propidium iodide (PI, 200 μg mL⁻¹ in 0.1% Triton X-100 containing 200 μg mL⁻¹ DNase-free RNase (Thermo Fischer Scientific EN0531) for 30 min at 37 °C. Following PI staining, cells were analysed by flow cytometry using an LSR Fortessa (Becton Dickinson) run by BD FACSDiva 8.0.1 software. The cell-cycle analysis was subsequently performed using FlowJo-V10 software (Tree Star Inc.). Cells were sorted into sub-G1, G1, S and G₂/M phase according to DNA content (PI signal). Quantification from gating on the respective histograms is shown as percent of live/singlet/PI-positive parent population per cell-cycle phase across different concentrations of the compound. Every experiment was performed in technical triplicates, at least three times independently, with a minimum of 10,000 (mean: 14,000) PI-positive singlet cells analysed per replicate.

**Immunofluorescence staining**. HeLa cells seeded on glass coverslips in 24-well plates (50,000 cells per well) were left to adhere for 24 h then treated for 24 h with **AzTax** under dark or lit conditions. Cover slips were washed then fixed with 0.5% glutaraldehyde, quenched with 0.1% NaBH₄, blocked with PBS + 10% FCS, treated with rabbit alpha-tubulin primary antibody (ab18251, abcam), washed, and incubated with donkey-anti-rabbit Alexa fluor 488 secondary antibody (A-21206, Invitrogen). After washing with PBS, coverslips were mounted onto glass slides using Roti-Mount FluorCare DAPI (Roth) and imaged with a Leica SP8 confocal microscope with a 63 × glycerol objective (DAPI: 405 nm, tubulin: 488 nm). Z-stacks (step size: 0.33 μm) were projected using Fiji and gamma values adjusted for visualisation.

**EB3 imaging with cell-specific photoisomerisation**. HeLa cells were transiently transfected with EB3-tdTomato (gift from Erik Dent; Addgene #50708) using FuGENE 6 (Promega) according to manufacturer's instructions (see Supplementary Information for all other imaging protocols). Cells were imaged on a Nikon Eclipse Ti microscope equipped with a perfect focus system (Nikon), a spinning disk-based confocal scanner unit (CSU-X1-A1, Yokogawa) and an Evolve 512 EMCCD camera (Photometrics) with a stage top incubator INUBG2E-ZILCS

(Tokai Hit) and lens heating calibrated for incubation at 37 °C with 5% $CO_2$. Cells were incubated in standard cell culture medium with 0.5% DMSO cosolvent, with or without *E*-AzTax3MP, for 10 min before microscope image acquisition using MetaMorph 7.7 was begun, with EB3-tdTomato imaging performed at 561 nm (0.17 mW, 300 ms every 4 s). Periods of intracellular-ROI-localised 405 nm illuminations (10 µW, 1 scan every 4 s during 24 s periods) were applied during imaging. Acquisition used a Plan Apo VC 60 × NA 1.4 oil objective. Comet count analysis was performed in ImageJ using the ComDet plugin (E. Katrukha, University of Utrecht, https://github.com/ekatrukha/ComDet).

**Subcellular MT manipulation in primary hippocampal neurons**. Mouse primary hippocampal neurons were derived from hippocampi of embryonic day 18 pups and plated on poly-L-lysine (Sigma-Aldrich) and laminin (Roche) coated coverslips. Primary neurons were cultured in Neurobasal medium (NB) supplemented with 2% B27 (Gibco), 0.5 mM glutamine (Gibco), 15.6 µM glutamate (Sigma-Aldrich) and 1% penicillin/streptomycin (Gibco) at 37 °C and 5% $CO_2$. Neurons were transfected with EB3-tdTomato on the 7th day in vitro (DIV 7) using Lipofectamine 2000 (Invitrogen) and were imaged at DIV 9. Neurons were imaged on the same system used for cell-specific-photoisomerisation experiments (described above). Neurons were first immersed in conditioned NB with 1% DMSO. EB3-tdTomato was imaged at 561 nm (0.1 mW, 400 ms every 4 s), while a ROI (violet box) was pulsed with 405 nm light (0.2 mW, 8 ms per trace) tracing over the ROI four times every 4 s with imaging frames interleaved. The same neurons were then immersed in conditioned NB with 1% DMSO and 0.5 µM AzTax3MP and immediately imaged for another 10 min; during this time the same ROI (violet box) was pulsed with 405 nm light (same protocol) starting 2 min into the acquisition.

**Statistics**. If not indicated otherwise all statistical analyses are unpaired, two-tailed Student's *t* test conducted with GraphPad Prism for Mac 8.4. *P* values below 0.05 were considered significant.

**Reporting summary**. Further information on research design is available in the Nature Research Reporting Summary linked to this article.

## Data availability

All data generated or analysed during this study are included in this published article and its Supplementary Information files, including the Source Data file provided (raw data for Figs. 2–5 and Supplementary Figs. 1–5). This and all data of the study can also be obtained from the authors upon request. None of these datasets are resources of public interest and therefore are not archived publicly in other forms. All materials are available from the Corresponding Author upon request. Source data are provided with this paper.

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

## Acknowledgements

This research was supported by funds from the German Research Foundation (DFG: SFB1032 Nanoagents for Spatiotemporal Control project B09 to D.T. and O.T.-S.; SFB TRR 152 project P24 number 239283807, Emmy Noether grant TH2231/1-1, and SPP 1926 project number 426018126 to O.T.-S.); the NIH (Grant R01GM126228 to D.T.; RO1AG050658 to F.B.); and the Thompson Family Foundation (TFFI to F.B.). We thank P.A.S. (LMU) for initial synthesis, F. Ermer and M. Borowiak (LMU) for initial MTT viability assays, H. Harz for microscopy access (LMU microscopy platform CALM), and CeNS (LMU) for support. We thank Natalia Marahori and Thomas Misgeld for valuable experimental feedback on **AzTax3MP**; and Maximilian Wranik, Michel Steinmetz, and the members of the Steinmetz group for work towards obtaining crystal structures of **AzTax**s bound to tubulin. We are indebted to Tim Mitchison for his contributions to small molecule inhibitors in MT research.

## Author contributions

A.M.-D. performed synthesis, photocharacterisation, and coordinated chemical data assembly. J.C.M.M. performed temporally reversible live cell imaging studies. K.L. and Y.K. performed cell biology. C.H. performed flow cytometry. R.B. performed in vitro tubulin polymerisation assays. K.I.J. and L.C.K. performed primary neuron isolation and culture. X.Q. and F.B. conducted early cell experimentation. A.A. supervised temporally reversible cell studies. J.A. performed and supervised cell biology, and coordinated biological data assembly. D.T. designed the concept and supervised initial synthesis. O.T.-S. designed the study, performed and supervised synthesis, supervised all other experiments, coordinated data assembly and wrote the paper with input from all authors.

## Funding

## Competing interests

The authors declare no competing interests.
