## [Peer Review File · Nature Communications]

Reviewers' comments:

Reviewer #1, an expert in microtubule biology (Remarks to the Author):

The authors describe the synthesis of a number of azobenzene-derivates of docetaxel to obtain a microtubule stabiliser with photoswitchable properties. Their best compound AzTax3MP has twofold higher biological activity in the "lit" state versus the "dark" state. It retains properties of docetaxel and promotes microtubule assembly in a bulk assay and slows microtubule polymerisation in cells, however, at significantly higher concentrations.

The main issue that limits usability of the compound is the moderate two-fold difference in activity between states, which results in cells being affected by the drug already before seeing any light.

The authors do not point this out, but if one analyses the microtubule polymerisation speed in their supplementary movies, a significant reduced microtubule assembly speed is found in the AzTax3MP-treated cells that are not illuminated. The drug-treated cells also show altered microtubule organisation even at the beginning of the movie, before any light was applied. Thus the present manuscript is at most a proof-of-principle that photoswitchable taxanes are possible to make, but I doubt that AzTax3MP is yet usable for the many applications the authors claim it is. Thus I don't expect that the paper will have a significant impact on the field, but rather describes a stepping stone towards an improved compound that might be developed in the future.

Specific points:

The manuscript is very wordy with especially the introduction being excessively long and repetitive, containing lengthy discussions of potential applications of photoswitchable microtubule stabilisers.

Some experiments are missing controls (e.g. TIRF imaging hasn't been shown for cells without light and without compound). Statistical analysis is missing from all data presented in the paper. Concentration of docetaxel in Fig. 2D needs to be given. Data in Fig. 3 aren't quantified at all, but text refers to substantial lower concentrations of AzTax being required to cause same effects in lit versus dark condition. Unsuitable references are used (for example ref 41 is inadequate to support a statement about EB3 being a nucleotide-dependent tip tracker). The authors overstate insights in the text, e.g. "... results demonstrate the use of AzTax3MP as a powerful tool ... with excellent spatiotemporal control" and later "such high-spatiotemporal-precision reagents". The authors don't even attempt subcellular activation of the drug or at least don't report any data on that. Other statements are just plainly wrong. Their compound cannot be used to induce local polymerisation or growth as their data in Fig. 5 show, quite contrary a reduction in assembly is evident already in the dark state and the number of comets highlighting growing microtubules is reduced upon illumination. I would also expect authors to discuss successful attempts of optogenetic modification of microtubules by photo-splittable or optical rerouting of microtubule regulators. The title doesn't make sense, the compound could be used for optical control, but won't control anything by itself.

Reviewer #2, an expert in azobenzene inhibitors (Remarks to the Author):

The authors outline a photo-switchable derivative of the microtubule stabilizing compound taxol. This work complements previous work of the authors and others on photo-switchable microtubule destabilizing compounds. The work is novel and high impact given the vast potential for such compounds in studies of microtubule dynamics, which play a substantial role in cellular trafficking, cancer, and neuronal development. Small molecule photo-switches have tremendous potential for studying and modulating cellular process, especially processes such as tubulin polymerization that cannot be easily controlled by optogenetic methods. The paper is not only important in the study of tubulin, it is also an important work in the field of photopharmacology because the authors present a detailed methodological road map for generation and testing of such compounds by systematically and creatively addressing many of the unique issues inherent to such systems such

as poor solubility and incomplete photoisomerization. Although the view of the authors that the disclosed blue shifted photoswitches are more desirable than red-shifted alternatives is highly optimistic given the limitations of blue and UV light *in vivo*, those limitations in no way detract from the significance of this advancement for photoswitch development and tubulin study.

The authors convincingly demonstrate that they have synthesized their desired compounds, characterized their photodynamic properties, and identified the source of the observed bioactivity to be tubulin stabilization. Importantly, the authors demonstrate the *in vitro* and *in cellulo* bioactivity of their molecules using multiple appropriate methods including microscopy, flow cytometry, and a tubulin polymerization assay. The statistical analyses appear generally appropriate for the types of methods employed and conclusions drawn. The work is well-reasoned, well-written, well-referenced and sufficiently well-detailed to be immanently reproducible. I highly recommend its publication in this journal.

Craig Streu

Reviewer #3, an expert in photochemistry and azobenzene chemistry (Remarks to the Author):

The manuscript describes the design, the synthesis of paclitaxel-azobenzenes with different substitution patterns in the azobenzene unit as photochromic agents to control microtubule stabilization kind of "remotely" by these chemical agents.

Although both by optogenetic methods and for example by using photoswitchable colchicines have been developed, no such stabilizers have yet been described.

The paper is not only very well written, it is very convincing from the rationale and design point of view, the medicinal chemical SARs applied to find photoswitchable compounds which differ in their biological activities, the photochemistry necessary, and especially the different biological readouts applied. The spatiotemporal control of MTs in cultured cells is particularly impressive. Also the use of the most recent data available like the cryo-EM structure of the paclitaxel-tubulin complex have been used.

The paper surely deserves publication, also in Nature Communications, since it not only by itself holds considerable interest, but it is a kind of a prime example how such photochromic compounds should be developed and what they can be applied for.

I would still suggest some revisions:

p. 2: The term "in vivo" applies to living organisms only. Even if this term has been used in a current review for all cellular systems, this is not correct. Only respective papers which truly use living organisms should be cited in this context (the number is fairly low actually).

p.3: probably the authors of ref. 34 are not particularly happy about the comments of the authors. Since this is of quite some relevance, the authors should either be less detailed or more detailed about commenting this manuscript.

p. 4: the p-amino compounds obviously thermally relax instantly, which could have been expected. The authors may discuss whether switching under constant irradiation might be useful in special setting instead of the photowitching in lipid media?

p. 6: the differences in cytotoxicity can be due to many, fairly diverse reasons. Albeit interesting, it is kind of speculative, and I recommend to shorten this part.

p.10: Of course, it is interesting to know what these compounds can be used for, but this part seems a bit too long unless this work has not been performed.

Dear Reviewers,

We are pleased to resubmit our manuscript ***Photoswitchable paclitaxel-based microtubule stabilisers optically control structure and dynamics of the microtubule cytoskeleton*** to *Nature Communications*.

Here we develop and show proof-of-principle applications for reagents that enable the efficient optical control of the endogenous microtubule cytoskeleton. These will greatly benefit the cell and chemical biology communities by solving a variety of problems that have until now prevented high-resolution cytoskeleton biology studies, and they may cast new light on the biology of the highly clinically important taxane drug class.

We are very grateful for the reviews and the editing process; thank you to all for your help and critique. We have now provided significant extra experiments, controls, and movies, supported by our expanded author list. These additional experiments, figures and movies include:

- new quantifications on live cell imaging (Fig S5a-b)
- new live cell imaging for MT density (Movies 3-4, Fig S5c)
- new subcellularly-resolved photoswitch applications in primary hippocampal neurons (Fig 6, Movies 5-8). These experiments particularly highlight both the spatiotemporal precision of these small molecule tools, as well as the application space that they may reach within neuroscience - one of the major goals for which we originally wished to develop these tools
- additional controls for movies previously reported (Movies 10-14)
- textual changes including adding background and discussion on optogenetics for the cytoskeleton; cutting a page of introduction; and diverse smaller edits to improve clarity.

For many of these we thank the reviewers for their suggestions and attention which motivated us to perform these improvements. We believe this revised manuscript will fully address all suggestions and concerns raised during the first review round and we are pleased to resubmit the new stronger paper.

We now respond item-by-item to comments raised.

Reviewer #1 an expert in microtubule biology:

The authors describe the synthesis of a number of azobenzene-derivates of docetaxel to obtain a microtubule stabiliser with photoswitchable properties. Their best compound AzTax3MP has twofold higher biological activity in the "lit" state versus the "dark" state. It retains properties of docetaxel and promotes microtubule assembly in a bulk assay and slows microtubule polymerisation in cells, however, at significantly higher concentrations.

The main issue that limits usability of the compound is the moderate two-fold difference in activity between states, which results in cells being affected by the drug already before seeing any light. The authors do not point this out, but if one analyses the microtubule polymerisation speed in their supplementary movies, a significant reduced microtubule assembly speed is found in the AzTax3MP-treated cells that are not illuminated. The drug-treated cells also show altered microtubule organisation even at the beginning of the movie, before any light was applied.

1. Further analysis of the effects of AzTax3MP without illumination were now performed (Fig S5) to quantify this since we do not want to avoid clearly discussing any features. Quantifications show a dark state reduction of EB3 comet velocity of about 20% (Fig S5a) but please note that the density of comets is not influenced (Fig S5b), and furthermore overall MT density in the cells is not affected (Fig S5c). While the 20% is statistically significant, it is in our opinion not a large number compared to the ca. 100% reduction after photoswitching, and we do discuss this (and e.g. its dose-dependency and potential for tuning) in the Main and the Supp Info (e.g. page S37-S38). We believe that as first-generation tools, the existence of background activity is to be expected and discussed, and given that photoactivation does present a drastic and above all useful change of properties, we think that the utility of the photoswitching is beyond doubt.

2. We also somewhat disagree with the number stated to summarise fold-change of activity. For example, Figure 2c's cell viability assays, on which the compound optimisation process was based, show a 5.8-fold activity change between the mostly-Z (360 nm) and all-E (dark) assays. In Fig S5b, the change of activity can also be read as 6-fold (if the density of polymerising MTs in the cell is of interest). Ultimately, the fold change seen depends on the experiment or readout used; and that the real fold change achievable has allowed us to apply useful photocontrol across a variety of cell types and systems. We outline too in the Discussion that AzTax3MP microscopy assays were performed at 405 nm which does not allow it to reach its full potential; but that the (*still unsolved*, but *hotly-pursued*...) challenge to make photoswitches with optimal 405-performance is probably set to improve this situation; this is part of our effort to extensively orient the field towards developing just such improved reagents as the reviewer would desire.

Thus the present manuscript is at most a proof-of-principle that photoswitchable taxanes are possible to make, but I doubt that AzTax3MP is yet usable for the many applications the authors claim it is. Thus I don't expect that the paper will have a significant impact on the field, but rather describes a stepping stone towards an improved compound that might be developed in the future.

1. We realised that to convince the reader we have to go a step further, so to demonstrate the utility of our compound in a more realistic application system we added major experiments demonstrating two of the critiqued points. We used **AzTax3MP** in primary neuronal cells to show (a) applications to this important cell type which is one of our goal systems, and (b) subcellularly localized activation by illuminating specific dendritic processes, which promises clear utility in a variety of neuroscience research applications. We truly believe that the tool principle of photoswitchable MT stabilisers should be capable of the applications we propose for them. Whether that means that **AzTax3MP** must itself achieve them all, or whether it is enough to show reasonable progress and applications on a hitherto unexplored concept while identifying parameters and ideas for improvement, must be left to scientific judgement and to the future. However, we do feel that this is the first compound of its kind, so a drawback (residual dark activity) can be expected, and note that optogenetic systems too have grown from somewhat inauspicious beginnings over many papers. We are optimistic for the ability of chemistry to likewise challenge accepted "limitations" on what is possible in biology.

Specific points: The manuscript is very wordy with especially the introduction being excessively long and repetitive, containing lengthy discussions of potential applications of photoswitchable microtubule stabilisers.

We integrated this with the opinions of the other reviewers concerning the writing and necessary deletions and additions. We hope that the scientific background and message are now clear.

Some experiments are missing controls (e.g. TIRF imaging hasn't been shown for cells without light and without compound). Statistical analysis is missing from all data presented in the paper. Concentration of docetaxel in Fig. 2D needs to be given. Data in Fig. 3 aren't quantified at all, but text refers to substantial lower concentrations of AzTax being required to cause same effects in lit versus dark condition. Unsuitable references are used (for example ref 41 is inadequate to support a statement about EB3 being a nucleotide-dependent tip tracker).

Controls are there now (new movies 10-14); concentration is given; statistics are given where appropriate (biological replicates; not for representative experiments which are given as technical replicates). A better reference is now given. It should now be clearer that, e.g., the effects seen under 360 nm at 0.5 μM are approximately seen for dark-incubated cells only around 1.5 μM . These controls and statistics strongly support the original arguments, so thank you for the sharp eye on details which allows the paper to be stronger.

The authors overstate insights in the text, e.g. "... results demonstrate the use of AzTax3MP as a powerful tool ... with excellent spatiotemporal control" and later "such high-spatiotemporal-precision reagents". The authors don't even attempt subcellular activation of the drug or at least don't report any data on that.

The temporal resolution had been discussed (eg. second-scale onset of inhibition in Fig 5) but should now be clearer. We thank the referee for the initiative to attempt and report spatial resolution, with satisfying subcellular activation (Fig 6). We hope that this establishes good reason to maintain our assessment of "high-spatiotemporal-precision", especially since there are no systems we are aware of that show similar performance (Introduction and Discussion).

Other statements are just plainly wrong. Their compound cannot be used to induce local polymerisation or growth as their data in Fig. 5 show, quite contrary a reduction in assembly is evident already in the dark state and the number of comets highlighting growing microtubules is reduced upon illumination.

The example concerned our incorrect use of the word "hyperpolymerisation" in two places. The two uses have been rewritten to better standards. They now read, "Firstly, after MT stabilisation is induced by AzTax E \rightarrow Z isomerisation then Z-AzTax:tubulin incorporation into the MT, the altered MT-dependent biology downstream of MT dynamics and network architecture probably cannot be instantaneously returned..." and "Spatially-selective induction of MT stabilisation will also be of particular interest for studies where local MT networks are thought to drive biology." We expect that this is acceptable. Beyond this we are unaware of other statements that are just plainly wrong, but would be happy to learn of and correct any that there are, and we hope that the previous incorrect usage does not more broadly cast doubt on the rest of the study.

I would also expect authors to discuss successful attempts of optogenetic modification of microtubules by photo-splittable or optical rerouting of microtubule regulators. The title doesn't make sense, the compound could be used for optical control, but won't control anything by itself.

Both are discussed more, within reasonable referencing limits, given that the focus of the paper is on chemical methods to directly address tubulin protein rather than genetic methods to operate on MTs through regulatory proteins. We like the title and want to keep it.

In conclusion we thank Reviewer 1, especially for the impetus to tackle subcellular photocontrol in an application that we had previously only claimed in the text, and we are pleased to have benefited from the close reading and high standards of her/his review.

Reviewer #2 an expert in azobenzene inhibitors:

The authors outline a photo-switchable derivative of the microtubule stabilizing compound taxol. This work complements previous work of the authors and others on photo-switchable microtubule destabilizing compounds. The work is novel and high impact given the vast potential for such compounds in studies of microtubule dynamics, which play a substantial role in cellular trafficking, cancer, and neuronal development. Small molecule photo-switches have tremendous potential for studying and modulating cellular process, especially processes such as tubulin polymerization that cannot be easily controlled by optogenetic methods.

We thank the reviewer for this assessment and opinion from the chemical side of the chemical biology community.

The paper is not only important in the study of tubulin, it is also an important work in the field of photopharmacology because the authors present a detailed methodological road map for generation and testing of such compounds by systematically and creatively addressing many of the unique issues inherent to such systems such as poor solubility and incomplete photoisomerization.

We are grateful for these remarks. Photopharmacology is indeed still young and we hope that the development story and the otherwise somewhat invisible details we go into (eg. the bis(hydroxyethyl)amide model switches, unfortunately consigned to the Supplementary Information due to space reasons but we think a solid feature) will bring an extensive positive impact to the field more generally.

Although the view of the authors that the disclosed blue shifted photoswitches are more desirable than red-shifted alternatives is highly optimistic given the limitations of blue and UV light in vivo, those limitations in no way detract from the significance of this advancement for photoswitch development and tubulin study.

We are happy to leave this as a topic for discussion perhaps even privately. But for microscopy reagents, we stand by the opinion that the bluer the better and we are happy to be the ones to launch this unconventional view in the photopharma arena; and we explain in better detail now to ground this view, "Therefore, maintaining orthogonality to the widest possible range of imaging wavelengths by blueshifting is in our opinion and experience more advantageous, since it can keep other imaging channels vacant for multiparameter, photoorthogonal studies". Of course for different applications, different properties are desirable, but even *in vivo* (multicellular embryonic or adult small animal systems) it can in some situations be an advantage to use the high absorption of UV plus the high isomerisation brightness of UV-photoswitching (compare to eg. diazocine >600 nm

extinction coefficients) to improve spatial precision while actually reducing net photon flux needed for switching.

The authors convincingly demonstrate that they have synthesized their desired compounds, characterized their photodynamic properties, and identified the source of the observed bioactivity to be tubulin stabilization. Importantly, the authors demonstrate the in vitro and in cellulo bioactivity of their molecules using multiple appropriate methods including microscopy, flow cytometry, and a tubulin polymerization assay. The statistical analyses appear generally appropriate for the types of methods employed and conclusions drawn. The work is well-reasoned, well-written, well-referenced and sufficiently well-detailed to be immanently reproducible. I highly recommend its publication in this journal.

Many thanks for the review and for the opportunity to highlight an important and we hope provocative opinion in the community.

Reviewer #3 an expert in photochemistry and azobenzene chemistry:

The manuscript describes the design, the synthesis of paclitaxel-azobenzenes with different substitution patterns in the azobenzene unit as photochromic agents to control microtubule stabilization kind of "remotely" by these chemical agents. Although both by optogenetic methods and for example by using photoswitchable colchicines have been developed, no such stabilizers have yet been described. The paper is not only very well written, it is very convincing from the rationale and design point of view, the medicinal chemical SARs applied to find photoswitchable compounds which differ in their biological activities, the photochemistry necessary, and especially the different biological readouts applied. The spatiotemporal control of MTs in cultured cells is particularly impressive. Also the use of the most recent data available like the cryo-EM structure of the paclitaxel-tubulin complex have been used. The paper surely deserves publication, also in Nature Communications, since it not only by itself holds considerable interest, but it is a kind of a prime example how such photochromic compounds should be developed and what they can be applied for.

Many thanks for the strongly positive review and for highlighting the several different aspects of our study, which we hope make it a strong interdisciplinary achievement.

I would still suggest some revisions:

p. 2: The term "in vivo" applies to living organisms only. Even if this term has been used in a current review for all cellular systems, this is not correct. Only respective papers which truly use living organisms should be cited in this context (the number is fairly low actually).

Indeed we intended this to mean in highly developed animals (if we are thinking of the same paper, then yes, the misuse of the term in a recent review is undisputed). To dispel any confusion we insisted on this explicitly, "in vivo in embryonic and adult animals".

p.3: probably the authors of ref. 34 are not particularly happy about the comments of the authors. Since this is of quite some relevance, the authors should either be less detailed or more detailed about commenting this manuscript.

Coincidentally, the article's authors recently published a followup paper implicitly showing that their effects were indeed independent of tubulin binding, but neither stated it nor retracted the original paper, and even continue to defend it. So after due process with the authors and then with the publisher, we published this critique in *Angewandte Chemie* (<http://dx.doi.org/10.1002/anie.201912616>) as a separate article and leave only 2

sentences commenting on it in this paper. Thanks for the suggestion to be as transparent as possible in the right medium.

p. 4: the p-amino compounds obviously thermally relax instantly, which could have been expected. The authors may discuss whether switching under constant irradiation might be useful in special setting instead of the photowitching in lipid media?

Agreed, it was expected. We don't believe in constant-irradiation switching in long-term biology since this can result in CALI-like protein deactivation (at least we think we have seen this experimentally with photoswitchable colchicine analogues) but it is off-topic for this paper and we won't venture deeper; however we do include a one-page discussion of fast-relaxing compounds' possibilities in the Supp Info (S26-S27) which might be of interest.

p. 6: the differences in cytotoxicity can be due to many, fairly diverse reasons. Albeit interesting, it is kind of speculative, and I recommend to shorten this part.

p.10: Of course, it is interesting to know what these compounds can be used for, but this part seems a bit too long unless this work has not been performed.

We increased the precision of language and pruned the discussion, we believe that it should now be acceptable.

Thank you for the review and we would also look forward to further discussions on the several topics that are clearly shared interests.

=====

We thank all reviewers and the editor for their time, effort, and their constructive questions and suggestions. We believe that most issues have now been clarified or solved, so we are happy to resubmit and we look forward to your feedback.

With best wishes,

Oliver Thorn-Seshold on behalf of all authors.

REVIEWER COMMENTS

Reviewer #1 (Remarks to the Author):

The revised manuscript contains some more data that clarify some of the issues I raised previously, but I still have a problem with the way the paper is presented. The graphical abstract suggests subcellular spatial control that seems not to be possible with the photoswitchable drug presumably because it diffuses too rapidly. Inconvenient data demonstrating the limitations of the approach are hidden in the supplement. Only outer halves of error bars are plotted in some graphs making differences look larger. I would request the authors to truthfully report both the promise and the limitations of the developed compound without the reader needing to dig into the details to understand those themselves.

The phrase "high spatiotemporal control" would to a cell biologist suggest seconds rather than minutes and hundreds of nanometer to few microns. The applications shown here suggest that rapid response is possible in some experiments, but spatial accuracy is severely limited due to the fast diffusion of the activated compound making it mainly applicable to cellular level and maybe individual neurites. Therefore, all statements claiming high spatiotemporal control should be amended.

It is not clear what the authors mean with their statement that tubulin is not amenable to direct functional control using genetics. Of course, it is. And in contrast to the statement in the introduction "their highly conserved, survival-critical nature does not permit function-affecting modifications to be viable", there are human patients with mutations in tubulin genes who are alive, but have symptoms that suggest the mutation in tubulin does affect microtubule function (e.g. PMIDs 20829227; 21292473; 23001566).

The introduction also explains that microtubule drugs "suppress all contemporaneous MT-dependent functions spatially indiscriminately. Therefore, they do not allow spatiotemporally precise MT inhibition on the length or time scales appropriate for selectively studying MT-dependent processes. This restricts their scope of applications and their utility for selective research into MT cytoskeleton biology." I note that this statement applies also to the compound developed in this study. I don't see how washing in the drug and then immediately illuminating the cells to activate the drug is much different from washing in docetaxel straightaway. The only benefit of the photo-switchable compound is that it is reversible. But nocodazole washout experiments are pretty commonplace to achieve that same effect. Why it is beneficial to have a docetaxel-like effect only in one cell in a dish rather than all of them, hasn't been made clear. But there is definitely no evidence that the compound can distinguish between different microtubule functions.

The paper makes some suggestions how this compound might enables new research avenues, for example to dissect how microtubule stabilisation facilitates neuronal regeneration following spinal cord injury. However, how this might be done is unclear. Considering the decay time, the damaged neuron versus surrounding tissue would need to be excited repeatedly and precisely over a relevant time period. How stabilising microtubules using a photoswitchable small molecule helps the study of intracellular transport is even more unclear.

Considering that AzTax3MP is hailed as one of its kind because it is a microtubule stabiliser, cellular readouts concern themselves solely with the inhibition of microtubule assembly. That effect can also be achieved by the microtubule destabilising photopharmaceuticals described previously. Thus, the study lacks any data or arguments which new aspect of microtubule biology is now accessible thanks to this compound.

Taken together, I am not impressed by the revised version. The authors obviously did not take my concerns too seriously and although other reviewers have been more positive from the outset, I note that they also have a different background and it is unsubstantiated statements and overselling the potential applications to cell biology that still cause concern.

Specific points:

- The authors might like the title, but it still doesn't make any sense. The new compound might

allow optically control, but does not optically control anything. For that it would need to emit light. How the structure of microtubules is affected by the compound is not shown either.

- Remove or redesign misleading graphical abstract.
- Delete high spatiotemporal control from abstract and throughout the manuscript.
- Remove statement on tubulin being not amenable to genetic control from abstract and introduction
- p.2 "eminent need", eminent means famous / notable. Not applicable here.
- p.2 citing a preprint (co-authored by the senior author) for a pretty generic statement on microtubule functions in the introduction is very poor practice. There are plenty suitable peer-reviewed primary research papers and reviews available to cite instead. I also don't think that citing the tubulin code paper here is a good choice.
- p.2 "Contemporaneous MT-dependent functions", contemporaneous refers to people living at the same time in history. Not applicable here.
- Fig 2b and 4c: show full errorbars or confidence intervals for both curves.
- Data shown in Fig.S5a should be included in main Figure 5. The graph in 5b can be compressed slightly to make space for the bar graphs showing microtubule assembly speed measurements.
- p.11 Please correct the sentence "Collectively, these results demonstrate the use of AzTax3MP as a powerful tool to directly optically modulate endogenous microtubule morphology, dynamics, and function in live cells with excellent spati- otemporal control." On page 9, the authors say that microtubule assembly, but not density is affected by AzTax3MP treatment and there are no data showing spatiotemporal control of microtubule morphology or function.
- p.12 Please correct the statement "as well as subcellularly-resolved inhibition in primary cells." "Subcellular" seems only be possible in mature neurons when illuminating separate neurites. Other primary cells will have the same limitations as the HeLa cells used elsewhere in the manuscript.
- The manuscript would benefit immensely from a section entitled "limitations" and a critical discussion of the limits to the applicability of the compound in its current form. This should include limits associated with the relatively narrow dynamic range, about two-fold effect of lit versus dark state, limits associated with the rapid diffusion of the compound limiting spatially precise manipulations and limits associated with the thus far unexplored properties of microtubule stabilised with the compound.

Reviewer #3 (Remarks to the Author):

The authors have addressed all issues raised by the reviewers and I think the paper is now publishable in its current form.

Reviewer #1 raised the point that the manuscript might be a proof-of-concept paper rather than a breakthrough. I think in this case it is more than a proof-of-concept due to the numerous applications provided and the compound optimization strategies.

But one can surely argue about that.

Dear Reviewers,

We are pleased to resubmit our manuscript ***Photoswitchable paclitaxel-based microtubule stabilisers allow optical control over the microtubule cytoskeleton*** to *Nature Communications*.

Here we develop and show proof-of-principle applications for "AzTax" photoswitchable microtubule-stabilising reagents, that allow the experimentalist to optically manipulate the polymerisation dynamics of the endogenous microtubule cytoskeleton. These reagents will benefit the cell and chemical biology communities by enabling spatiotemporally-resolved cytoskeleton biology studies, and they may cast new light on the biology of the clinically important taxane drug class.

We are very grateful for the second round of reviews; thank you for your helpful critique. We have now modified the text, changed the title, and rearranged figure panels to achieve a stronger paper. We thank the reviewers for their suggestions and believe this revised manuscript will fully address all points raised during the second review round.

We now respond item-by-item to comments raised.

Reviewer #1 an expert in microtubule biology:

The revised manuscript contains some more data that clarify some of the issues I raised previously, but I still have a problem with the way the paper is presented. The graphical abstract suggests subcellular spatial control that seems not to be possible with the photoswitchable drug presumably because it diffuses too rapidly. Inconvenient data demonstrating the limitations of the approach are hidden in the supplement. Only outer halves of error bars are plotted in some graphs making differences look larger. I would request the authors to truthfully report both the promise and the limitations of the developed compound without the reader needing to dig into the details to understand those themselves.

We have removed the central panel of the graphical abstract to avoid any potential for misunderstanding. The abstract text has been reworked for clarity and to avoid any interpretation as overstating results. We have moved some more information from the SI to the main text (see Fig 5, discussed below) though we do not see this data as inconvenient since it supports the analysis we present in the paper; the decision to place it (referenced and not hidden) in the Supporting Information had been made to avoid mixing data from two different experiments conducted with two different purposes in mind. The unidirectional error bars that we had used to reduce clutter in two graph panels have been replaced by bidirectional bars (Fig 2b, Fig 4c). We do not consider that we distort information and we extensively address Limitations in the Discussion section. The point-by-point response to those Limitations is given below (last section).

The phrase “high spatiotemporal control” would to a cell biologist suggest seconds rather than minutes and hundreds of nanometer to few microns. The applications shown here suggest that rapid response is possible in some experiments, but spatial accuracy is severely limited due to the fast diffusion of the activated compound making it mainly applicable to cellular level and maybe individual neurites. Therefore, all statements claiming high spatiotemporal control should be amended.

Our goal has been to highlight the increase of spatiotemporal precision possible with our photoswitchable stabilisers as compared to what is possible with wash-in/wash-out experiments. We feel that second-level temporal precision is well-established by our experiments. We agree that we have shown micron-level control only in the neuronal application. We have removed every mention of "high spatiotemporal precision" and replaced some by the phrasing from the abstract, e.g. "temporal precision on the scale of seconds and spatial precision on the level of tens of microns" (e.g. at Fig 5). This is now unambiguous.

It is not clear what the authors mean with their statement that tubulin is not amenable to direct functional control using genetics. Of course, it is. And in contrast to the statement in the introduction “their highly conserved, survival-critical nature does not permit function-affecting modifications to be viable”, there are human patients with mutations in tubulin genes who are alive, but have symptoms that suggest the mutation in tubulin does affect microtubule function (e.g. PMIDs 20829227; 21292473; 23001566).

We now rephrased to read "However, while optogenetics has succeeded in providing motors and scaffold-associated proteins that are responsive to externally controlled stimuli, no **optogenetic variants of the basic cytoskeleton scaffold proteins actin and tubulin** have been achieved. An exogenously-controllable system for directly patterning cytoskeleton scaffold dynamics and structure with spatiotemporal resolution would however be highly desirable, since it would allow researchers to modulate any of the cytoskeleton-dependent functions." The new version is now unambiguous that we are talking about **optogenetics** only, which had been the original goal (old version "one **salient feature of optogenetics for cytoskeletal control** remains that no genetic methods have yet allowed **direct spatiotemporally-resolved control** over the polymer organisation and dynamics of the scaffold proteins (actin, tubulin etc) themselves, probably since their highly conserved, survival-critical nature does not permit function-affecting modifications to be viable.")

For the abstract, it was impossible for us to rephrase our concept to similar clarity while staying within the word limit so we deleted the clause which referenced tubulin.

The introduction also explains that microtubule drugs “suppress all contemporaneous MT-dependent functions spatially indiscriminately. Therefore, they do not allow spatiotemporally precise MT inhibition on the length or time scales appropriate for selectively studying MT-dependent processes. This restricts their scope of applications and their utility for selective research into MT cytoskeleton biology.” I note that this statement applies also to the compound developed in this study. I don’t see how washing in the drug and then immediately illuminating the cells to activate the drug is much different from washing in docetaxel straightaway. The only benefit of the photo-switchable compound is that it is reversible. But nocodazole washout experiments are pretty commonplace to achieve that same effect. Why it is beneficial to have a docetaxel-like effect only in one cell in a dish rather than all of them, hasn’t been made clear. But there is definitely no evidence that the compound can distinguish between different microtubule functions.

The introduction is a statement of a conceptual challenge facing all non-responsive inhibitors. The concept of photoswitchable inhibitors is indeed that photoswitchability

enables spatial and temporal specificity. Conceptually, already several studies have shown the power of being able to perform cell-resolved manipulations e.g. within a developing embryo and these were received with some interest by the community (see Zenker *et al.* Science 2017 and Cell 2018, which we reference). Those studies would never be possible with wash-in/wash-out; and such is the target audience for our reagents.

It is not suggested that this or any small molecule can distinguish different MT functions. What is said is for example "AzTax also inhibit **MT-dependent functions** such as successful completion of mitosis", which we show by experiment.

The paper makes some suggestions how this compound might enable new research avenues, for example to dissect how microtubule stabilisation facilitates neuronal regeneration following spinal cord injury. However, how this might be done is unclear. Considering the decay time, the damaged neuron versus surrounding tissue would need to be excited repeatedly and precisely over a relevant time period. How stabilising microtubules using a photoswitchable small molecule helps the study of intracellular transport is even more unclear. Considering that AzTax3MP is hailed as one of its kind because it is a microtubule stabiliser, cellular readouts concern themselves solely with the inhibition of microtubule assembly. That effect can also be achieved by the microtubule destabilising photopharmaceuticals described previously. Thus, the study lacks any data or arguments which new aspect of microtubule biology is now accessible thanks to this compound. Taken together, I am not impressed by the revised version. The authors obviously did not take my concerns too seriously and although other reviewers have been more positive from the outset, I note that they also have a different background and it is unsubstantiated statements and overselling the potential applications to cell biology that still cause concern.

Ours is a proof of concept study and we feel we present it as such. We also intend it to stimulate thinking about further generations of ideas and tools (hence we give a stepwise description of the development and validation process) as well as thinking about the new possibilities that could be addressed with the introduction of photoswitchability onto scaffolds of interest. Therefore we think that our general and forward-looking statements, such as "this first demonstration of photoswitchable MT stabilisers **is a substantial step towards** high-spatiotemporal-precision studies of a range of critical processes in cell biology," are appropriately pitched.

We cite why stabilisers have made impacts on research, we state that there are no previous photoswitchable MT stabilisers, we show that AzTax are photoswitchable MT stabilisers. We explain how conceptually the advance of photoswitching over wash-in/wash-out corresponds to the advance of optogenetics over transfection and we cite how various photoswitchable reagents have indeed made impacts on research through their photoswitchability enabling previously impossible studies. Whether this AzTax reagent or any future photoswitchable stabilisers will ever make any impact on future research (i.e. are actually good tools) can only be shown after biological users try it out, or after chemists are induced to make something better; we certainly believe that they both will.

We state for the reader why we do not feel that AzTax are functionally equivalent to photoswitchable destabilisers; as we say in the Introduction, "While both MT destabilisers and MT stabilisers can be used to suppress microtubule polymerisation dynamics in cell culture, stabilisers have enabled a variety of research and human therapeutic applications which are inaccessible to destabilisers, due to their differing pharmacology, stoichiometry, and spectrum of biological effects," and we cite several examples of studies and effects which it has only been possible to achieve using stabilisers (epothilones, taxanes) and not colchicinoids. *In vivo* direct antitumor activity in the cancer context is another example.

We also feel that through the provision of new experiments, statistics and heavy editing in the previous round, as well as numerous reviewer-driven changes, we have been taking the critique and assessment of all the reviewers seriously throughout the review process.

#Specific points:

- *The authors might like the title, but it still doesn't make any sense. The new compound might allow optically control, but does not optically control anything. For that it would need to emit light. How the structure of microtubules is affected by the compound is not shown either.*

Changed.

- *Remove or redesign misleading graphical abstract.*

Done, see above.

- *Delete high spatiotemporal control from abstract and throughout the manuscript.*

Done. The retained "highly spatiotemporally resolved" we have reserved for optogenetics, although to be honest it is questionable if even that method would pass the "hundreds of nanometer" criterion therefore we feel the 100 nm distinction is a little arbitrary.

- *Remove statement on tubulin being not amenable to genetic control from abstract and introduction*

Done, see above.

- *p.2 citing a preprint (co-authored by the senior author) for a pretty generic statement on microtubule functions in the introduction is very poor practice. There are plenty suitable peer-reviewed primary research papers and reviews available to cite instead. I also don't think that citing the tubulin code paper here is a good choice.*

Correct. The generic transport and motility statement is now referenced to a textbook instead of the tubulin code paper. The Kopf paper (published) is still a well-adapted reference for discussing morphological plasticity.

- *Fig 2b and 4c: show full errorbars or confidence intervals for both curves.*

Done, see above.

- *Data shown in Fig.S5a should be included in main Figure 5. The graph in 5b can be compressed slightly to make space for the bar graphs showing microtubule assembly speed measurements.*

This requested to mix data from two different types of experiments; it is not clear to us that it improves the paper to do so, but it is done nevertheless.

- *p.2 "eminent need", eminent means famous / notable. Not applicable here.*
- *p.2 "Contemporaneous MT-dependent functions", contemporaneous refers to people living at the same time in history. Not applicable here.*

This was a specific request. Googling for dictionary definitions, both did seem allowed as used; nonetheless both are now changed.

• p.11 Please correct the sentence “Collectively, these results demonstrate the use of **AzTax3MP** as a powerful tool to directly optically modulate endogenous microtubule morphology, dynamics, and function in live cells with excellent spatiotemporal control.”

Amended, “demonstrate the use of **AzTax3MP** as a powerful tool to directly optically modulate endogenous microtubule **network architecture**, **polymerisation** dynamics, and **MT-dependent** functions in live cells”.

• p.12 Please correct the statement “as well as subcellularly-resolved inhibition in primary cells.” “Subcellular” seems only be possible in mature neurons when illuminating separate neurites. Other primary cells will have the same limitations as the HeLa cells used elsewhere in the manuscript.

Amended for clarity.

• The manuscript would benefit immensely from a section entitled “limitations” and a critical discussion of the limits to the applicability of the compound in its current form. This should include limits associated with the relatively narrow dynamic range, about two-fold effect of lit versus dark state, limits associated with the rapid diffusion of the compound limiting spatially precise manipulations and limits associated with the thus far unexplored properties of microtubule stabilised with the compound.

The limitations to our approach and the most important avenues for further development are discussed in the Discussion section.

The point about stabilised-MT properties is correct, thank you for pointing it out. We address it with a new paragraph “From a biological perspective, while **AzTax** have reproduced the effects expected for taxanes across these studies, comparing their mechanism more deeply remains to be addressed...” The limitation of diffusion is a fact of life for all small molecule reagents i.e. a limit of the field of chemistry itself (although photoswitchable reagents operated under dynamic photoswitching suffer far less serious diffusive limitations to their spatiotemporal precision as compared to washin/washout reagents...); we instead present a new paragraph within the small molecule space, beginning “Reducing the halflife of spontaneous relaxation of an **AzTax** to its less-binding isomer could also offer perspectives for improving the ease...”

The other limitation points raised had previously been presented and we now added the word “limitation” to signal it clearer (“If **AzTax** reagents with still greater bioactivity differentials could be accessed, this would improve reagent performance with respect to several limitations of the current best candidate **AzTax3MP**.”). (1) The most important limitation remains conceptual – once stabilised, abnormally organised MTs are formed, instantaneous off-switching of the compound would not be able to instantaneously restore normal function (“there is probably a limit to the temporal resolution of *true* biological reversibility that any photoswitchable stabiliser can display...”). (2) A wider dynamic range would of course make the reagents even better (paragraph “differential bioactivity”) and this is discussed openly, both from points of view of isomeric (“we do not believe that improving the completeness of bidirectional isomeric photoswitchability...”) as well as bioactivity photoswitching (“would improve reagent performance with respect to several limitations... dynamic range...”) More importantly, we again outline the strategies we feel will be most likely to address these limitations.

In this way we feel we communicate a range of chemical design limitations, strategies that will improve them, and show in what areas these improvements will impact biological users. As to the applicability of the concept of a photoswitchable stabiliser *per se*, given these discussions of current room for improvements we think there should not be too great limitations put on the reader’s imagination for the future.

We thank Reviewer 1 for their close reading and the high standards of her/his review.

Reviewer #3 an expert in photochemistry and azobenzene chemistry:

The authors have addressed all issues raised by the reviewers and I think the paper is now publishable in its current form. Reviewer #1 raised the point that the manuscript might be a proof-of-concept paper rather than a breakthrough. I think in this case it is more than a proof-of-concept due to the numerous applications provided and the compound optimization strategies. But one can surely argue about that.

We thank Reviewer 3 for their time and their opinion.

=====

We thank all reviewers and the editor again for their time, effort, and their constructive questions and suggestions. We believe that all issues have now been clarified or solved, so we are happy to resubmit and we look forward to your feedback.

With best wishes,

Oliver Thorn-Seshold on behalf of all authors.